

# Beyond binary baseflow separation: delayed flow index as a fresh perspective on streamflow contributions

Michael Stoelzle[1], Tobias Schuetz[2], Markus Weiler[1], Kerstin Stahl[1], and Lena M. Tallaksen[3]

[1]Faculty of Environment and Natural Resources, University Freiburg, Germany
[2]Department of Hydrology, Faculty VI Regional and Environmental Sciences, University of Trier, Germany
[3]Department of Geosciences, University of Oslo, Norway

*Correspondence to*: Michael Stoelzle (michael.stoelzle@hydro.uni-freiburg.de)

**Abstract.** Understanding components of the total streamflow is important to assess the ecological functioning of rivers. Binary or two-component separation of streamflow into a quick- and slow (often referred to as baseflow) component, and the
associated and often used baseflow index (BFI), have been criticised for their arbitrary choice of separation parameters. These methods also merge different delayed components in one baseflow component. As streamflow generation during dry weather often results from drainage of multiple sources, we propose a novel delayed flow index (DFI) considering the dynamics of multiple delayed contributions to streamflow. The DFI is based on characteristic delay curves where the identification of breakpoint estimates helps to avoid rather arbitrary separation parameters and allows distinguishing four types of delayed
streamflow contributions. The methodology is illustrated using streamflow records from a set of 60 headwater catchments in Germany and Switzerland covering a pronounced elevational gradient of roughly 3000 m. We found that the quickflow signal often diminishes earlier than given by BFI-analyses with only two flow components, and further distinguished a variety of flow contributions with delays shorter than 60 days controlling the seasonal streamflow dynamics. For streamflow contributions with delays longer than 60 days, the method was used to assess catchments' water sustainability during dry
spells. Colwells's Predictability, a measure of streamflow periodicity and sustainability, was applied to attribute the identified delay patterns to streamflow dynamics and catchment storages. The smallest storages were consistently found for catchments between approx. 800 and 1800 m a.s.l. Above an elevation of 1800 m the DFI suggests that seasonal snowpack provides the primary contribution, whereas below 800 m groundwater resources are the major streamflow contributions. Our analysis also indicates that subsurface storage in high alpine catchments may be large and is overall not smaller than in lowland catchments.
The DFI can be used to assess the range of sources forming catchments' storages and judge their long-term sustainability of streamflow. Combining the DFI with a low flow stability index, allows us to better understand controls of seasonal low flow regimes across different regimes.

## 1 Introduction

Sustained freshwater availability is essential for ecosystem functioning. During dry weather sustained streamflow is important
for groundwater-surface-water-interactions (Sophocleous, 2002), streamflow drought severity (Zaidman et al., 2002), the





variability of water temperature (Constantz, 1998) or the dilution of contaminants (Schuetz et al., 2016). In hydrology, sustained streamflow and hence freshwater availability, is often estimated by the amount or timing of baseflow or as a single-value baseflow index (BFI). The BFI is the proportion of baseflow to total streamflow, i.e. higher BFI values are interpreted as an indicator of more water being provided from stored sources (Tallaksen and van Lanen, 2004). Total streamflow is

composed of quick- and baseflow. Quickflow is the portion of total streamflow originating rather directly from precipitation input (also termed direct runoff or stormflow). In contrast, different names for baseflow (i.e. sustained flow, delayed flow, groundwater flow, dry- or fair-weather flow) highlight its relevance during prolonged dry weather. Baseflow has commonly been considered "as the portion of flow that comes from groundwater storage or other delayed sources" (Hall, 1968), i.e. water that has previously infiltrated into the soil and recharged to aquifers, but can also origin from other sources of delayed flow.

In later studies it is described as the contribution from persistent and slowly varying sources, including groundwater flow (Sophocleous, 2002). Dingman (2015) understands baseflow as water maintaining streamflow between water-input events. Between these events different sources such as groundwater, melt water from snow, glacier or ice, lakes, riverbanks, floodplains, wetlands, spring or return flow from irrigation can contribute to the baseflow component of streamflow (Smakhtin, 2001). Considering these definitions, the amount of baseflow and its seasonal variability is controlled by multiple delayed

sources. Groundwater contributes to the baseflow component of river's streamflow, but baseflow is not a measure of groundwater storage only.

Baseflow is often estimated using hydrometric- or tracer-based hydrograph separation to decompose the different streamflow contributions (Smakhtin, 2001). Hydrometric-based hydrograph separation has a long history, but has been also criticized for ambiguous results compared to newer approaches based on chemical- or isotopic tracers (Klaus and McDonnell, 2013). The

latter approaches are assumed to be physically more meaningful and allow assessing the water age (transit and residence time modeling), the mixing of the water (e.g. pre-event and event water) and the sources of different water contributions (e.g. groundwater, snowmelt). However, isotope or chemical data sets are often only available for a short period. Hence, hydrograph analysis is often the only possibility to provide an estimate of the amount of slowly varying flow, i.e. delayed contributions, to streamflow.

Quantification and attribution of the delayed responses to potential sources are important to predict streamflow, including its seasonality and water sustainability. Seasonal consideration of delayed contributions to streamflow is particularly important for environmental flow assessment as both high flow and low flow variability are crucial for the integrity of river ecosystems (Acreman, 2016). Different delayed contributions may sustain streamflow during periods with low precipitation and are thus important to assess the vulnerability of aquatic ecosystems to climate change (e.g. Olden et al., 2011). During wet conditions,

streamflow contributions will have predominantly short delays (e.g. due to frequent rainfall events). During dry conditions catchment-specific drainage of water storages, such as groundwater aquifers, snowmelt, wetlands etc., control the stability and the interannual variability of low flows. The spatio-temporal variability of delayed contributions to streamflow cause major concerns regarding reliability and resilience of hydropower generation, low-flow forecast, water supply or water management,





especially in water or data scare areas with high water demand (e.g. due to irrigation) (e.g. Griffiths and Clausen, 1997;
Smakhtin, 2001; Tallaksen, 1995).

Regarding the terminology, quick- and baseflow might be inconclusive as sometimes the term quickflow is seen as a
description of the response, whereas the term baseflow is seen as a description of the source. For this reason, it has been
suggested to separate the flow into a fast- and slow component, taking a purely response-orientated perspective (Kronholm
and Capel, 2015). Then the quick- and baseflow terminology related to event- and pre-event water, takes a more source-
orientated perspective. Kronholm and Capel (2015) introduced a Slow Flow Index to "separate water that is moving slowly
through the hydrological system" and consequently concluded that baseflow (defined as groundwater discharge by the authors)
is not always the same as slowflow. However, fast- and slowflow might be also an insufficient terminology or classification
as all contributions to streamflow have specific delays: some are slow, but others are even slower. In the proposed fast-slow
flow concept, slowflow will, as well as the source-orientated baseflow, include contributions from multiple sources; but neither
different response signals nor the contributions from different sources are distinguishable. Freyberg et al. (2018) therefore
recommend developing hydrograph separation beyond the traditional separation of pre-event and event water to identify
eventually many different sources of streamflow.

In the past, two-component baseflow separation methods such as the recursive digital filtering (Lyne and Hollwick, 1979;
Nathan and McMahon, 1992) or separation based on progressively identified streamflow minima in the IH-UK (Institute of
Hydrology, United Kingdom) baseflow separation method (Gustard et al., 1992; Natural Environment Research Council,
1980), have proven a simple and practical way of indexing the catchment response. Both methods were developed in regional
studies (e.g. Australia, United Kingdom) and need reasonable, but subjective, decisions on parameterization to separate quick-
and baseflow from total flow. The proposed parameter ranges reflect region-specific streamflow response characteristics (e.g.
for BFI the choice of 5-days windows for separation in the UK is adapted to the typical rainfall regime in the UK) and would
have to be recalibrated for other climates as demonstrated e.g. for seasonal snow regimes by (Tallaksen, 1987) or for
intermittent stream by (Aksoy et al., 2008). However, allowing general applicability to catchments containing a range of
different delayed sources, each with possibly different origins and varying contributions, another approach might be more
appropriate to disentangle delayed components in the BFI-"baseflow" component (Halford and Mayer, 2000).

To tackle some of these issues we extend common hydrograph analysis (i.e. baseflow separation) of a binary quick- and
baseflow separation into an analysis considering multiple delayed contributions to streamflow. The delayed flow index (DFI)
is tested for a set of catchments in Germany and Switzerland covering a pronounced elevational gradient acting as a proxy for
different streamflow regimes. From lowland to montane to alpine catchments catchment characteristics generally show gradual
changes. Overall, lowland catchments are thought to have thicker soils, larger groundwater storages and longer growing season.
Montane catchments comprise pronounced slopes, large elevation ranges and higher freeze-thaw-dynamics due to high
variations in catchment snowpack. Alpine catchments are often snowmelt- or (occasionally) glaciermelt-dominated, they have
thinner soils and are characterized by bedrock, gravel and taluses, and are near or above the treeline. Given the variety of
ecosystems along elevational gradients, we can expect that catchment characteristics are also reflected in various streamflow



regimes (e.g. rainfall- or snowmelt-dominated catchments with low flows in different seasons) and that multiple delayed contributions with specific signals (e.g. stormflow, snowmelt or groundwater contributions) are distinguishable. As catchment
storage is both seasonally stored surface water (e.g. snow) as well as sub-surface water stored with less pronounced seasonality (e.g. deep groundwater aquifers), we used Colwell's Predictability as a thorough metric to assess streamflow predictability considering both facets of water storage in catchments. The objectives of our study are:

1)   to develop a delayed flow separation method with the ability to quantify multiple delayed streamflow contributions and

2)   to evaluate the reliability and applicability of this method by linking delayed flow contributions to catchment characteristics, storage and low flow variability.

## 2 Methods

### 2.1 Delayed flow separation method

This study is build on the widely used IH-UK baseflow separation method (Gustard et al., 1992), also referred to as the
smoothed minima method. The IH-UK method was developed for humid, rainfall-dominated catchments in the United Kingdom (UK) and separates the total flow into two components (quick- and baseflow), i.e. above and below a baseflow hydrograph derived from a daily streamflow series of perennial streams. For a thorough description of the original method see e.g. Hisdal et al. (2004) and the Manual on Low-flow Estimation and Prediction of WMO (2009). This IH-UK method identifies local minima in daily streamflow series and a continuous baseflow hydrograph is obtained by linear interpolation
between the identified local streamflow minima. The separation method is based on identifying streamflow minima in consecutive periods of $N$=5 days (block size) and a multiplying factor $f$ (also referred to as a turning point parameter) that determines whether the minimum is identified as a local minimum or not ($f$ equal 0.9 in the original method). The estimated baseflow hydrograph is more sensitive to changes in parameter $N$ than to changes in the turning point parameter $f$ (Aksoy et al., 2008; Tallaksen, 1987). Hence, we focus in this study only on the variation of block size $N$, which can be seen as an
estimate of an average streamflow delay and catchments response (i.e. unit of $N$ is 'days').

It has further been suggested to calculate the BFI separately for different seasons using different $N$ values to avoid identifying minima during seasons with a different runoff response (to that of rainfall), such as spring flood due to snowmelt (Tallaksen, 1987). Aksoy et al. (2008; 2009) adapted the IH-UK method for perennial and intermittent streams accounting for the sensitivity of BFI to different block sizes $N$. They also compared the IH-UK method to other filter separation methods such as
the recursive digital filter method (Lyne and Hollwick, 1979), and were amongst others aware of the sensitivity of BFI to different block sizes N (Miller et al., 2015; Piggott et al., 2005). However, to our knowledge, a comprehensive analysis of the sensitivity in BFI to different block sizes $\underline{N}$ is still missing. Aksoy et. al (2008) suggested to determine catchment-specific values for $N$ as a function of catchment area $A$ [km$^2$] with $N = 1.6A^{0.2}$ , however, if applied as "a rule of thumb" (Linsley et al., 1958),  $N$ will only vary between roughly 2 and 10 days for catchment areas between 10 and 10.000 km$^2$. Thus, there is a





need for a more systematic approach. In this study, we expand on the IH-UK method (i.e. smoothed minima method) to derive
a delayed flow index ($DFI_N$) for integer values of $N$ ranging between 1 and 90 days, as follows:

1.   Similar to the BFI procedure (WMO, 2009) the time series is divided into non-overlapping consecutive blocks of $N$ days.

2.   The minimum value of each block is compared to the minimum of the two adjacent blocks. If a factor $f = 0.9$ times

of the minimum value, is less than or equal to the two adjacent minima, a turning point (TP) is defined. TPs will be
separated by a varying number of days due to the algorithm.

3.   The TPs are connected by straight lines to become the delayed flow hydrograph. Between TPs the delayed flow
values are derived by linear interpolation. If the estimated delayed flow exceeds the original streamflow value, the
delayed flow is replaced by the original streamflow value.

4.   The delayed flow index for a given $N$ ($DFI_N$) is then calculated as the ratio of the sum of delayed flow to the sum of
total streamflow.

$N = 0$ represent the case of no separation and the delayed flow series is set equal to the streamflow series ($DFI_0 = 1$). For
$N = 1$ the DFI value will be slightly less than 1 as some peaks in the hydrograph will be cut by the $f = 0.9$ criterion. The BFI
value of the original method is equal to $DFI_5$, i.e. $N = 5$ days. Theoretically, $DFI_N$ (as well as the original BFI value) range

between 0 and 1. With increasing $N$ the length of each consecutive period increases and DFI decreases because TPs are set
wider apart and more and more streamflow peaks (i.e. contributions with shorter delays) are excluded from the separation as
illustrated in Figure 1. Here the methodology is demonstrated for three catchments with different streamflow regimes (Fig.
1a and 1b), showing the full range of delayed contributions as a continuous change from $N = 1$ (only the sharpest peaks are
identified) to $N = 60$ (all peaks are separated). With increasing $N$ the $DFI_N$ shows a monotonic decrease and converges to a

catchment-specific limiting value for large values of $N$ (Fig. 1c). $DFI_N$ would only approach 0, if streamflow series has
regularly zero flow periods (intermittent streams): Zero flow must then occur approximately every $N$ days, which was not the
case for any of our study catchments.

As an appropriate maximum block size ($N_{max}$) is unknown a priori, we originally calculated the $DFI_N$ index for block sizes
from 1 to 180 days (cf. Sect. 2.2) providing characteristic delay curves (CDC) characterizing the relationship between block

size $N$ and $DFI_N$. $DFI_N$ values and resulting CDCs are calculated for the the whole year and separately for the summer season
(May to October) and the winter season (November to April) to allow the seasonal variability of different contributing
sources to be assessed. The final CDC was smoothed by choosing the minimum of two consecutive DFI values for all pairs
of $DFI_i$ and $DFI_{i+1}$.



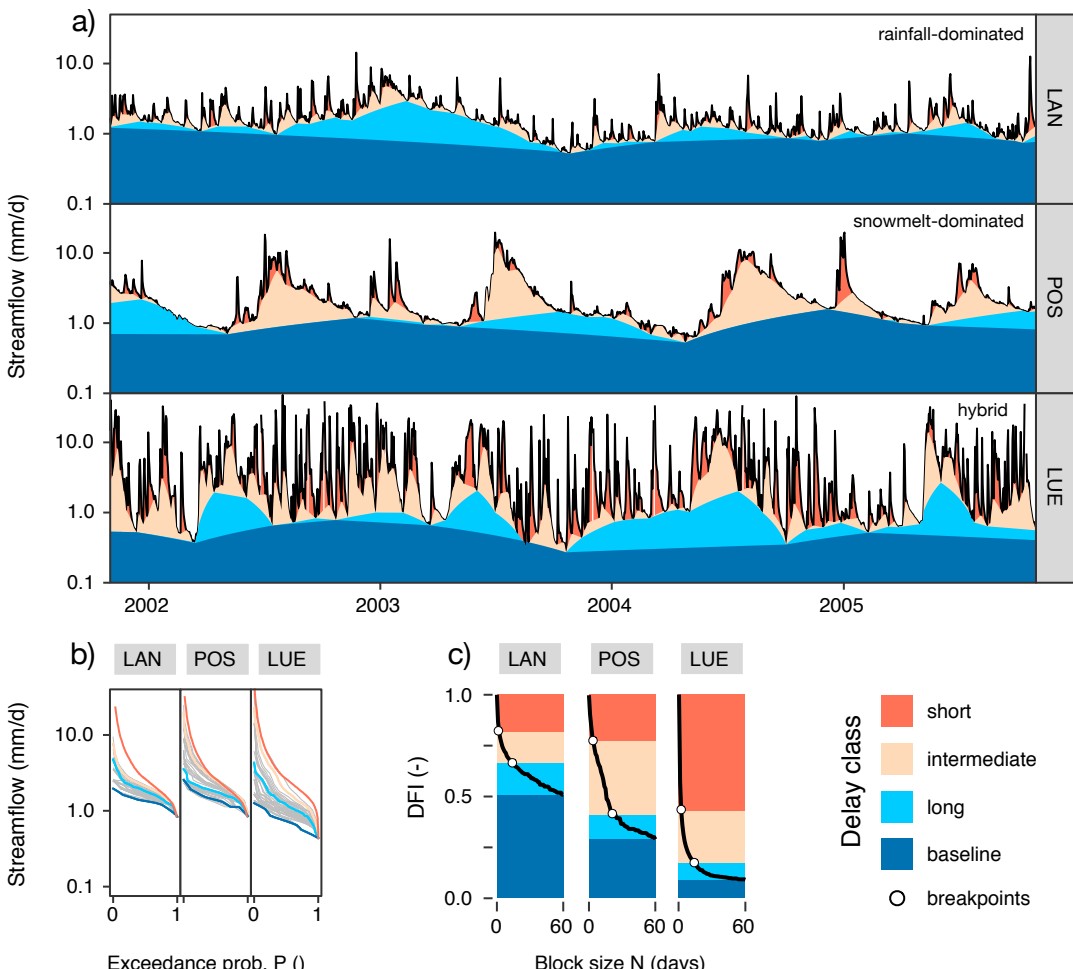

**Figure 1: a) Delayed flow separation for three catchments from Switzerland, namely Langeten (LAN), Poschiavino (POS) and Lümpenenbach (LUE), with different streamflow regimes, b) Flow duration curves for all separated delayed flow hydrographs (1-60 days) and c) DFI$_N$-values for different block size N shown in combination with breakpoints (circles). Colours refer to the four different delay classes identified (Section 2.3). Note the logarithmic y-axis in a) and b).**

## 2.2 Maximum block size N$_{max}$

Some studies (Wahl and Wahl, 1995) have shown that CDCs converge to a catchment-specific asymptotic value for large $N$. Accordingly, we hypothesize that for a given $N_{max}$ the proportion of delayed streamflow stays nearly constant even if $N$ is further increased ($N > N_{max}$). This value, which is considered a typical (maximum) delay of all contributing sources, is then captured by this maximum block size $N_{max}$. In the seasonal low flow period, streamflow typically originates from one or only a few delayed sources (e.g. slowly draining groundwater aquifers). We thus attributed the variability of the annual minimum





streamflow (AM) to the slowest contributing source(s) and identified $N_{max}$ by comparing the fraction of mean annual minimum flows (MAM) to mean streamflow (MQ) as an indicator of low flow sensitivity to $DFI_N$ values for block sizes $N$ between 2 and 180 days. This implies that $N$ in the delayed flow separation is increased until a clear relationship between MAM/MQ values and $DFI_N$ values for all catchments and N values is established. The relationships between MAM/MQ and $DFI_N$ is

shown in Figure 2 for different $N$. As the block size $N$ is increasing, the maximum block size $N_{max}$ is identified as the point when the explanatory power of the regression between MAM/MQ and $DFI_N$, the coefficient of determination (Fig. 2, insets), ceases to increase. Based on this initial analysis, $N_{max}$ is set to 60 days as $N = 60$ is sufficient to capture all annual minimum flows across the catchments and larger values of $N$ provide no additional information on streamflow variability (i.e. CDCs flatten out for $N > 60$, cf. Sect. 2.3).

**2.3 Breakpoints and delay classes**

Figure 3 shows a general decrease in DFI values with increasing $N$, but the rate of decrease varies among the catchments. It is assumed that a change in slope of the CDC indicates a transition from faster to slower contributing sources (stores) in the catchment. Such specific values of $N$ can be defined as breakpoints (BP) splitting the CDCs into piecewise linear segments with different slopes (Miller et al., 2015; Wahl and Wahl, 1995). We calculated two breakpoints between 0 and 60 days for

each CDC by minimizing the residual sums of the resulting three linear regressions (Muggeo, 2008). Accordingly, the linear regression represents the piecewise linear shape of the CDCs for the four segments as shown in Figure 3 for four random catchments from the data set (catchment A, B, C and D). The position of each breakpoint pair ($DFI_{BP1}$ and $DFI_{BP2}$; given as integer values) together with the associated $DFI_{60}$ value, hence characterize the shape (e.g. curvature, changes in slope, minimum level) of each single CDC. The delayed contributions to streamflow are then classified into four delay classes, and

quantified as the ratio of each component to the total annual streamflow (ranging between zero and one) for each class (Fig. 3):

- short delay ($D_S$): between $N = 0$ (equal to original streamflow series) and $BP_1$
- intermediate delay ($D_I$): between the two breakpoints ($BP_1$ and $BP_2$)
- long delay ($D_L$): between the $BP_2$ and $N = 60$

- baseline ($D_B$) delay: equals the $DFI_{60}$ value ($N = 60$).

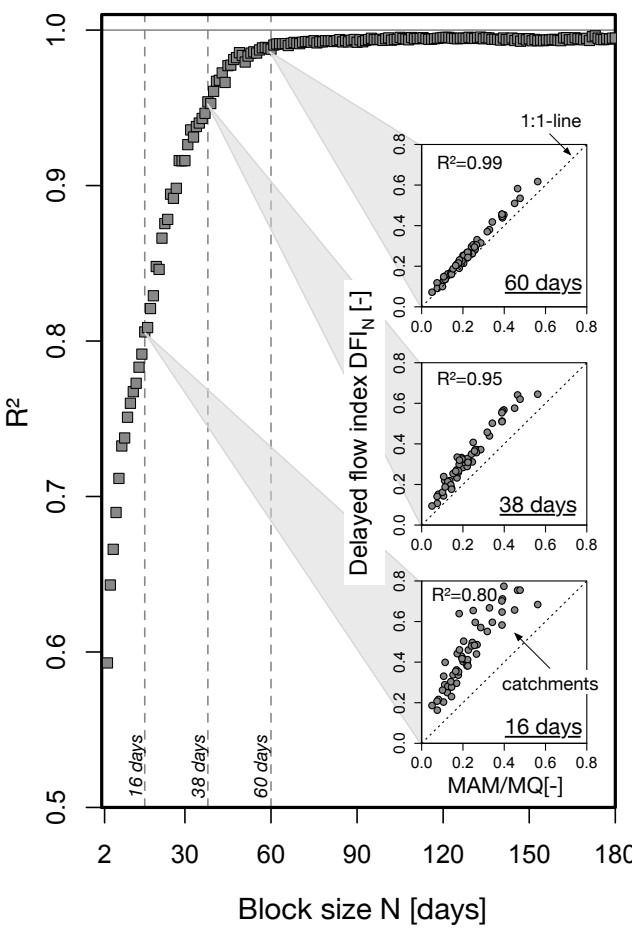

**Figure 2: The Coefficient of determination, $R^2$, between $DFI_N$ and the ratio of Mean Annual Minimum flow to Mean Flow (MAM/MQ) for varying block size N ranging between 2 and 180 days. Insets show the degree of agreement (as compared to a 1:1 line) for N = 16, 38 and 60 days.**

The resulting four delay classes can be interpreted according to their relative contributions to streamflow, but also in terms of their absolute values (e.g. mean annual water volume contributing to streamflow in each delay class). Absolute streamflow contributions in each delay class are then calculated based on the catchment-specific average annual streamflow. Relative contributions are calculated based on the differences of the DFI values, i.e. relative contribution in the delay class $D_S = DFI_0 - DFI_{BP1}$, $D_I = DFI_{BP1}-DFI_{BP2}$, $D_L = DFI_{BP2}-DFI_{60}$, and $D_B = DFI_{60}$.

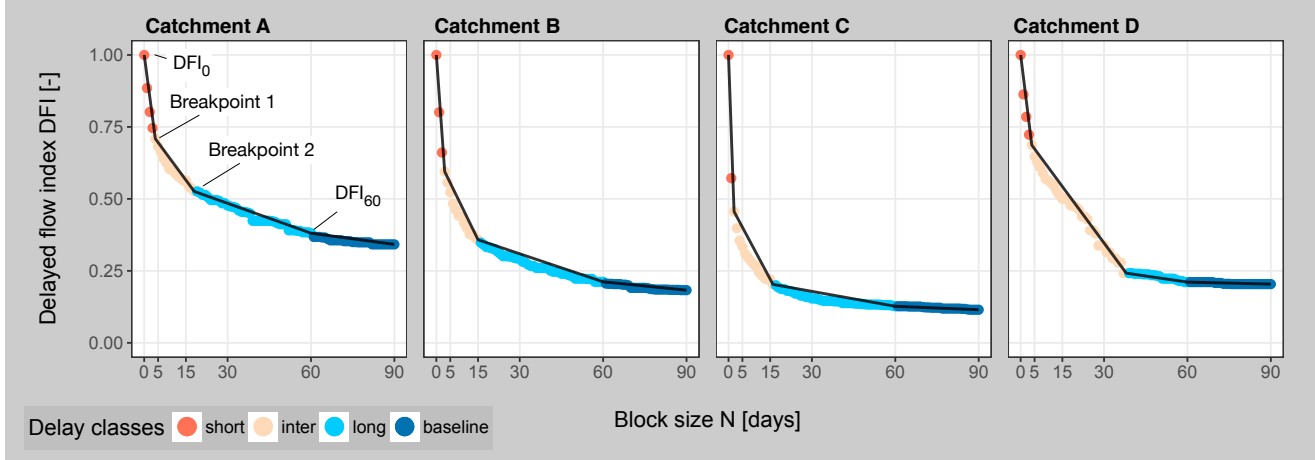

**Figure 3: Various CDC curves for four example catchments A-D and their variation in DFI$_0$, DFI$_{60}$ and breakpoints 1 and 2. The example catchments are extracted from the study data set to highlight the variety of CDCs and ratios of delayed streamflow contributions.**

### 2.5 Colwell's Predictability

One aim of the study is to assess to what degree it is possible to attribute the varying delayed flow contributions to catchments storages (and processes). To answer this question, we utilize Colwell's Predictability (Colwell, 1974) to classify different delay classes of streamflow based on the predictability, constancy, and seasonality of streamflow regimes. Colwell's Predictability is an approach to compare regime constancy or stability in multi-catchment studies with pronounced elevational gradients and different streamflow regimes (Viviroli and Weingartner, 2004). Colwell developed a metric to assess the uncertainty of periodically changing environmental variables with respect to state and time. The detailed mathematical derivation of Colwell's Predictability can be found in Colwell (1974). Examples of applied Colwell's Predictability are the periodicity of fruiting and flowering (Colwell, 1974) or the analyses of streamflow patterns (Poff, 1996) and precipitation (Gan et al., 1991). Based on information theory, the uncertainty of a variable with respect to its state and timing has been defined as an estimate of reciprocal predictability. This means, if the state and/or timing of a variable is highly uncertain, it is also poorly predictable. This, in turn, leads to a highly predictable flow regime when flow is nearly invariant throughout the year (state is known) or when streamflow has a clear year-to-year seasonal pattern (timing is known). Combining variation in state and timing the total Colwell's Predictability $P_T$ [-] is calculated as

$$P_T = P_c + P_S$$

with a component for constancy $P_C$ [-] and a component for contingency or seasonality $P_S$ [-]. All three values can vary between 0 and 1 under the condition $P_C + P_S \leq 1$. The variables $P_C$ and $P_S$ are calculated with the *R* package hydrostats with standard configuration (Bond, 2016). A value of $P_T = 1$ indicates that the mean monthly streamflow values show the same temporal





pattern (here streamflow regime) for each temporal cycle (here the hydrological year) (Gan et al., 1991). If so, Constancy $P_C$ is 1 (e.g. constant flow without any seasonality) or Seasonality $P_S$ is 1 (e.g. pronounced seasonality with identical monthly flows from year-to-year) or $P_C$ and $P_S$ add theoretically up to 1. In reality, smaller values for $P_T$ are found due to the variability of climate and the influences of catchment characteristics and water uses (i.e. stronger year-to-year variability).

**2.6 Sensitivity in delayed flow components during low flow**

Here we investigate the different streamflow contributions and delays for months with low streamflow. The low flow stability index $S_{LF}$ is calculated for each calendar month $m$ based on 37 years of streamflow data (1976-2012) for ten randomly selected catchments in each catchment group (RLWR, RUPR, HYBR, SNOW) with:

$$S_{LF(m)} = \bar{Q}_{l4,m} / \bar{Q}_m,$$


where $\bar{Q}_{l4,m}$ is the average streamflow of the four driest months (~10% of data) and $\bar{Q}_m$ is the average monthly streamflow of all 37 years. $S_{LF(Apr)}$ is, for example, the average of the four driest Aprils in the time series divided by the average streamflow in April. The average of the four driest months is considered to represent the variability in the ~10 % most extreme low flow months and preferred to using only one month with the lowest streamflow. The low flow stability index $S_{LF}$ varies between 0

and 1 and quantifies how stable streamflow is during the low flow season. High $S_{LF}$ suggest relatively stable flows during low flow seasons, lower values indicate pronounced variability in the low flows compared to the monthly mean flow. A seasonal low flow sensitivity analysis is undertaken including the months January, April, August and October to account for different low flow generating processes dominating in different streamflow regimes (i.e. winter versus summer low flows).

**3 Data and regime classification**

We use daily streamflow data (1976-2012) with flow rates per unit area (mm d$^{-1}$) for a set of 60 headwater catchments with catchment areas between 0.54 and 955 km$^2$, all located in south-western Germany and Switzerland (Fig. 4). Mean catchment elevations range from 227 to 2377 m a.s.l., whereas maximum catchment elevation ranges from 338 up to 3231 m a.s.l. Some of the high-elevation catchments include small proportions of glaciers (2 - 7%). Although most of the catchments are not pristine, human interventions in these headwater catchments are often negligible and streamflow is assumed to be mostly near-

natural. A few streamflow records are influenced by hydropower (i.e. hydropeaking) or sewage discharge; however, the interventions are considered minor.

Mountain regions are heterogeneous in many aspects (morphology, geology, climate, etc.). Since their catchment characteristics offer many plausible catchment classifications, we classify catchments with a rather simple, but straightforward scheme based on the mean and maximum catchment elevation, reflecting hydrological regime types: rainfall-dominated (mean





catchment elevation below 1000 m a.s.l.), snowmelt-dominated (mean catchment elevation above 1600 m a.s.l.) and "hybrid"-regime catchments (i.e. mixture of rainfall and snowmelt) between these elevation bounds (Table 1, Fig. 4).

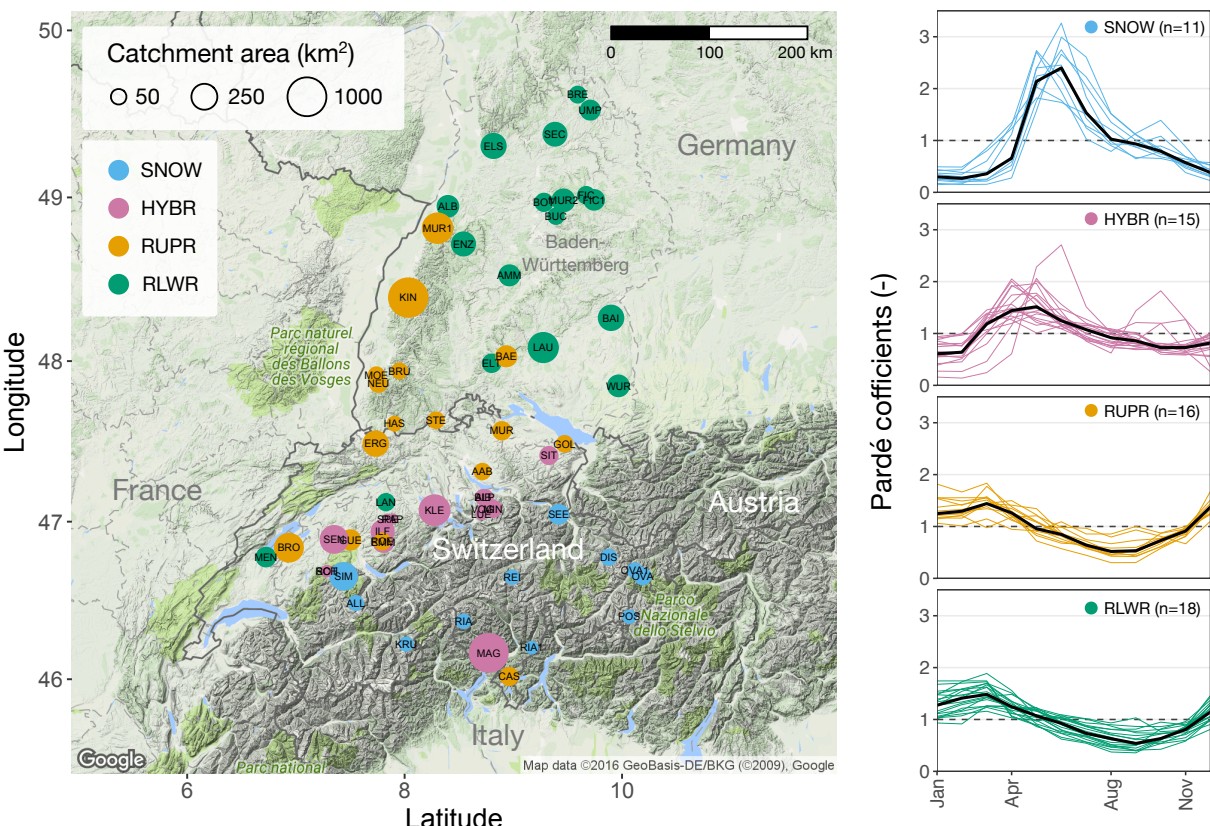

**Figure 4: Location and area (size of circle) of study catchments. Catchment classification (colors) is explained in Table 1. The right panel shows Pardé coefficients, i.e. the ratio of the long-term mean monthly streamflow to the long-term mean annual streamflow**
**for all catchments grouped by regime type.**

The classification follows the definitions of mountains and lowlands for the Alps by Viviroli and Weingartner (2004), which are Pardé coefficients to characterize the seasonality of streamflow (e.g. different typical low flow periods, Table 1). Rainfall-dominated catchments were further divided into "lower" and "upper" catchments with maximum elevations below and above
1000 m a.s.l. to consider differences in seasonal snowmelt and evaporative processes between the groups. The number of classes and lower and upper elevation bounds are comparable to classifications used in other studies in the same region (Jenicek et al., 2015; Staudinger et al., 2017; 2015; Viviroli and Weingartner, 2004).





**Table 1: Classification scheme separating the catchments into four different classes (abbreviation and color coded) according to elevation and hydrological regime types. Typical low flow periods are derived from streamflow data. Information on snow onset and snowmelt periods are derived from literature (Klein et al., 2016) and generalized for the regime types HYBR and SNOW.**

| Regime type | Classification scheme | | Mean Elevation (m a.s.l.) | Maximum Elevation (m a.s.l.) | Typical low flow period | Typical snow onset | Typical begin of snowmelt |
| | Class | Color code | | | | | |
|---|---|---|---|---|---|---|---|
| **rainfall-dominated** (lower elevation) | RLWR |  | < 1000 | < 1000 | Aug - Sep | variable | variable |
| **rainfall-dominated** (upper elevation) | RUPR |  | < 1000 | > 1000 | Aug - Sep | variable | variable |
| **rainfall- and snowmelt** | HYBR |  | 1000 - 1600 | - | Jan - Feb | Nov - Dec | Mar - Apr |
| **snowmelt-dominated** | SNOW |  | > 1600 | - | Jan - Mar | Oct - Nov | Apr - May |

Our study catchments are uniformly distributed over the four classes allowing for a balanced statistical analysis across the four regime types (Table 2). The catchment characteristics and hydrometeorological metrics show a decrease in catchment area $A$ with elevation, whereas precipitation $P$, streamflow $Q$ and the runoff ratio $Q/P$ are generally increasing with elevation.

**Table 2: Catchment characteristics and hydrometeorological metrics (based on period 1992-2013) of the four catchment groups. Numbers given are the average values along with the standard deviation (in brackets) of catchments within one group.**

| Class | Number of catchments (- / %) | Catchment Area $A$ (km$^2$) | Mean Elevation (m a.s.l.) | Precipitation $P$ (mm a$^{-1}$) | Streamflow $Q$ (mm a$^{-1}$) | $Q/P$ (-) |
|---|---|---|---|---|---|---|
| RLWR | 18 (30) | 152 ± 102 | 512 ± 181 | 1038 ± 191 | 387 ± 141 | 0.36 ± 0.07 |
| RUPR | 16 (27) | 172 ± 253 | 767 ± 158 | 1433 ± 234 | 732 ± 236 | 0.50 ± 0.09 |
| HYBR | 15 (25) | 152 ± 257 | 1213 ± 166 | 1803 ± 200 | 1240 ± 397 | 0.67 ± 0.17 |
| SNOW | 11 (18) | 19 ± 8 | 2025 ± 254 | 1635 ± 187 | 1529 ± 358 | 0.92 ± 0.13 |

## 4. Results

### 4.1 Characteristic Delay Curves

The CDCs demonstrate a high variability among catchments and within catchments groups. In Figure 5a the CDCs of all catchments are shown grouped by regime type and season (summer and winter), whereas the average CDCs for each regime type is shown in Figure 5b (averages per season). In general, the shape of the CDCs for rainfall-dominated (lower elev.) (RLWR) catchments decreases more slowly for increasing $N$ than is the case for rainfall-dominated (upper elev.) RUPR and rainfall- and snowmelt (hybrid) HYBR catchments. The shapes of the CDCs and also the values of $DFI_{60}$, (indicated by the boxplots in Fig. 5a), vary markedly among all catchment groups. Steeper CDCs imply higher streamflow dynamics, whereas





a gentle decrease in CDCs indicates a higher ratio of longer delayed contributions (compare Fig. 1). Seasonal differences in CDCs suggest different streamflow generation processes, e.g. in snowmelt-dominated SNOW catchments rather stable winter flows and higher flashiness during summer flows dominates. We found a higher variation in the CDCs in RLWR and SNOW

catchments (interquartile range: IQR) of $DFI_{60}$ between 0.12 and 0.20 for all seasons) than in RUPR and HYBR catchments (IQR of $DFI_{60}$ between 0.06 and 0.12 for all seasons). In RUPR and HYBR, the CDCs have a smaller range and show a faster decrease compared to RLWR and SNOW catchments. Overall, CDCs have a small or near-zero slope for delays $N > 60$ (Fig. 5a), however, some CDCs continue to decrease, although slowly, for $N > 60$. The relative changes in DFI between $N = 60$ and $N = 90$ are in all cases relatively small compared to the changes for the range $0 < N < 60$. The proportion of $DFI_{60}$-$DFI_{90}$ to

$DFI_0$-$DFI_{60}$ is overall small, varying between 6% (RLWR) and 1.5% (HYBR) with an average of 3% for all catchments.

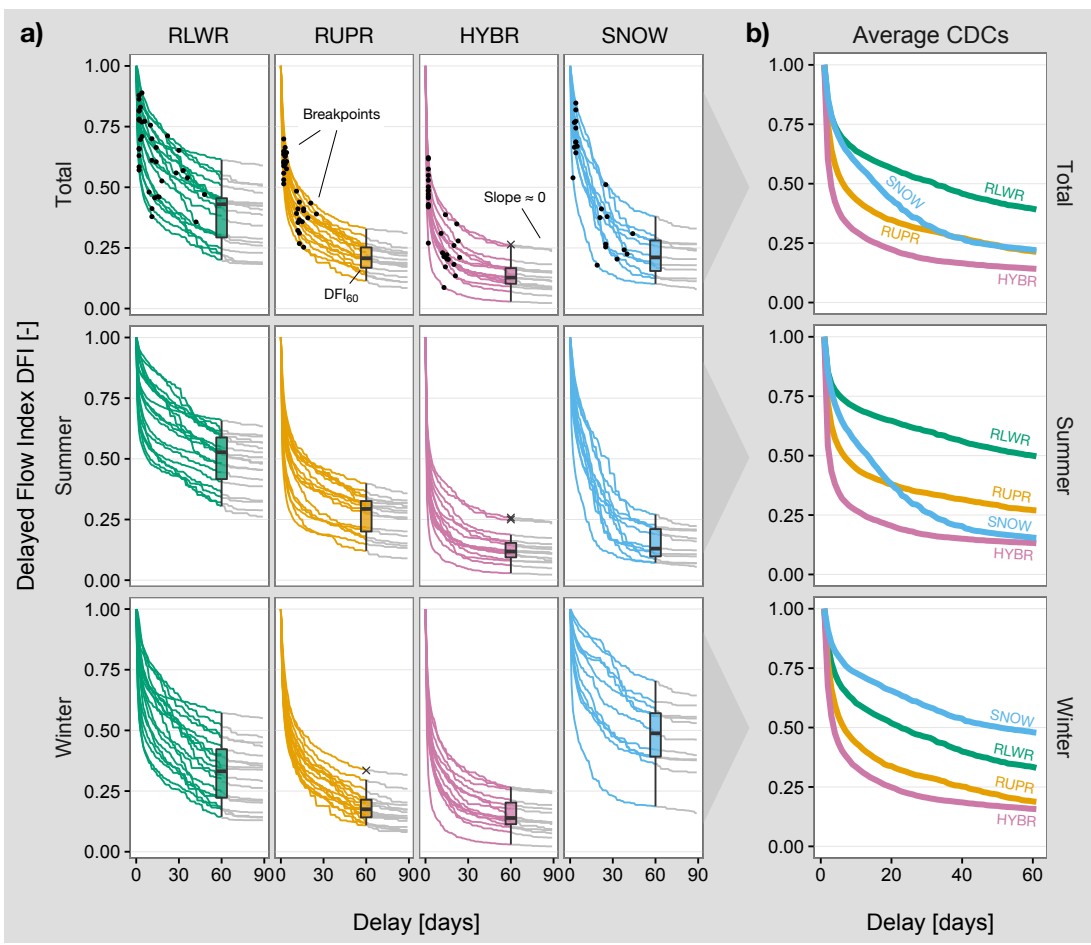

**Figure 5:** Characteristic delay curves (CDCs) for a) all catchments and b) as average for each catchment groups. CDCs are shown for the whole year, and summer (May-Oct) and winter (Nov-Apr) separately. Black dots are estimated breakpoints. Boxplots show

the distribution of $DFI_{60}$ values. The grey lines (delay > 60 days) have very small or zero slopes.





From a hydrological perspective, the $DFI_{60}$ (baseline delay $D_B$; Section 2.4) value is important as it quantifies the streamflow contribution of slowly varying sources with delays of 60 days and longer. Considering the whole year, RLWR catchments have on average the largest proportion of slowly varying sources (0.39), whereas SNOW (0.22), RUPR (0.21) and HYBR catchments (0.14) have notable lower $DFI_{60}$-contributions (Fig. 5). Compared to the annual analysis, the summer season (May-

Oct) $DFI_{60}$ is higher for RLWR and RUPR catchments (+10% and +5%) and lower for HYBR and SNOW catchments (-1% and -6%). For the winter season (Nov-Apr) average $DFI_{60}$ are lower compared to the whole year for RLWR and RUPR catchments (-6% and -3%) and higher for SNOW catchments (+26%) and almost equal for HYBR catchments (+1%). This result reveals that low-elevation catchments have relatively fewer streamflow contributions with longer delays during winter (e.g. due to the snow season and melt events), whereas SNOW catchments show higher streamflow contributions with longer

delays during winter low flows (Fig. 5b). The HYBR catchments have the smallest $DFI_{60}$ values for both seasons and the corresponding CDCs are characterized by a rapid decrease as $N$ increases until a value of approximately N = 20-30 days where the curve flattens in both summer and winter. In case of HYBR catchments (Fig. 5a), we found that on average 65% of the streamflow contributions have delays of 5-days or less ($DFI_5$ = 0.35), 78% have 20-days or less ($DFI_{20}$ = 0.22) and 84% have 40-days or less ($DFI_{40}$ = 0.16).

**4.2 Breakpoint estimates and streamflow contributions in delay classes**

The locations of the first and second breakpoints ($BP_1$ and $BP_2$) show some distinct features for the four catchment groups. The breakpoint estimates for RLWR and SNOW catchments are generally further apart than for RUPR and HYBR catchments. Short delayed contributions corresponding to $BP_1$ are between 2 and 4 days for 95% of the catchments. Three catchments have $BP_1$ of 5, 6 or 10 days, respectively. $BP_2$ are around 15 days for RUPR and HYBR catchments and around 25 days for RLWR

and SNOW catchments, indicating that the RLWR and SNOW catchments have overall larger streamflow contributions with intermediate delays.

Transforming the resulting CDC fractions into delay classes many study catchments show overall larger streamflow contributions from the short delay class ($D_S$), second largest contributions in the intermediate delay ($D_I$) or the baseline delay ($D_B$) and the smallest streamflow contributions in the long delay class ($D_L$). This suggests that $D_S$ contributions are often

important for streamflow generation. However, the DFI-analysis unveils also exceptions from this pattern with dominant streamflow contributions in $D_B$ for RLWR catchments and in $D_I$ for SNOW catchments (Fig. 6a, lower panel). These dominant contributions account for around 40% of the total streamflow in both catchment groups (Fig 6b, lower panel) and are clearly larger than the $D_L$ contributions in these groups. In contrast, HYBR catchments have an average $D_S$ contribution of 50% and an average $D_S$ plus $D_I$ contribution of 75%. For all HYBR catchments, the first breakpoint is consistently assigned at a delay

of $N = 2$ (Fig. 5a), highlighting the importance of fast streamflow generation processes and comparable fast streamflow recessions. Beside $D_S$ also $D_I$ contributions stand out and show a clear increase in absolute streamflow contributions with increasing elevation (see also Sect. 4.3, Fig. 7).

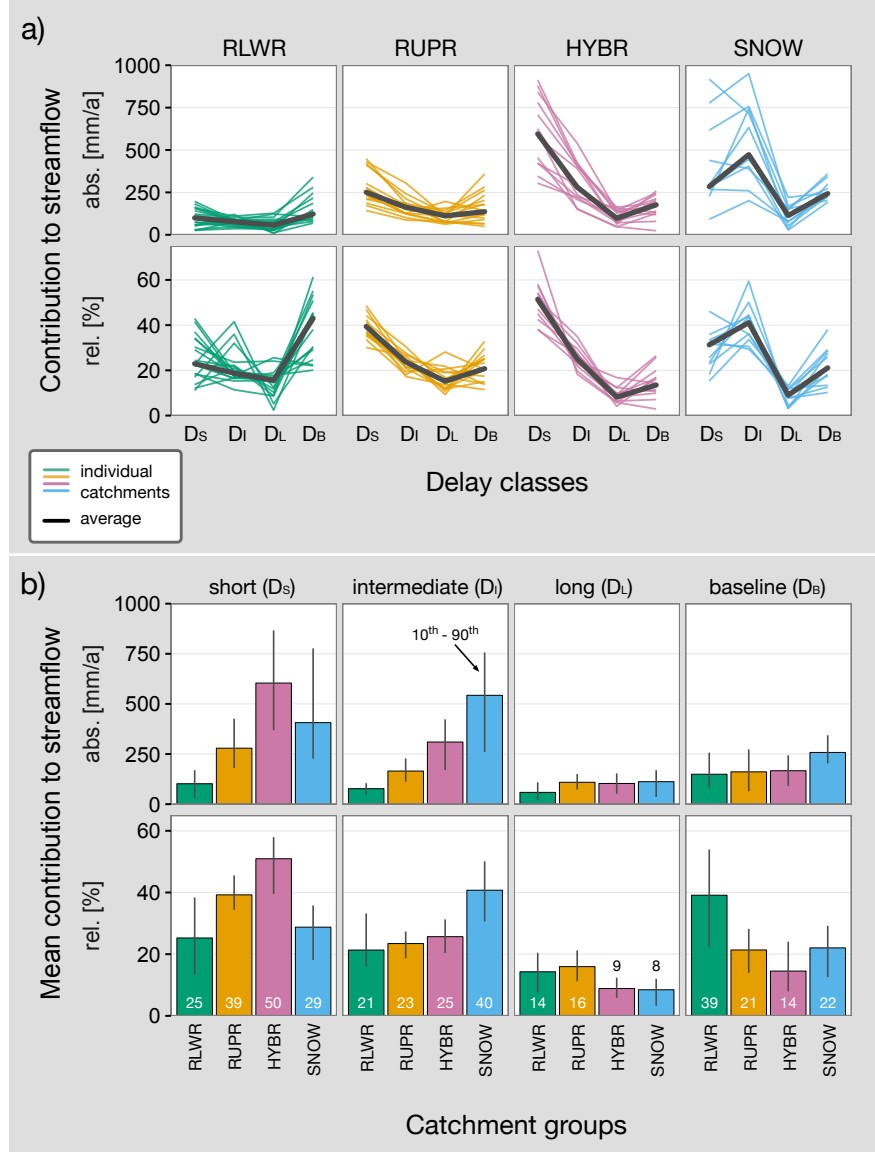

**Figure 6: Delayed contributions (in absolute and relative terms) to streamflow according to a) delay classes and b) catchment groups. In panel a) each coloured line intentionally represents one catchment to highlight the catchment-specific composition of different contributions.**

### 4.3 Elevational patterns of delayed flows

To explore the elevation-dependent pattern of delayed contributions in more detail and to investigate whether these results are sensitive to the catchment classification scheme (Table 1), we sorted the catchments by the mean catchment elevation and binned ten catchments together to calculate smoothed relative streamflow contributions for the four delay classes as shown in Figure 7. This analysis reveals distinct patterns of varying streamflow contributions. Below approximately 800 m a.s.l the





contributions for all delay classes show a high variability and the delay classes are less distinguishable. Above this elevation, the different delay classes show a clear pattern. $D_S$ contributions dominate in an elevation range between approx. 800 and 1800 m a.s.l.. Below 800 m a.s.l. $D_L$ contributions dominates whereas above 1800 m a.s.l. $D_I$ contributions are more prominent. The

peak of the $D_S$ contribution is around 1300 m a.s.l. corresponding to the smallest $D_B$ contribution. $D_L$ contributions decrease with increasing elevation levelling off at around 10% streamflow contribution slightly above 1500 m a.s.l. $D_B$ contributions are large for a few low-elevation catchments and show an opposed pattern to $D_S$ contributions. The decreasing $D_S$ contributions for elevations higher than 1300 m a.s.l. are compensated not only by $D_I$, but also by $D_B$ contributions. Catchments above 2000 m a.s.l. have larger $D_I$ than $D_S$ contributions and $D_B$ contributions are almost as large as $D_S$ indicating that at these elevations

intermediate and baseline delayed contributions control around 60% of the streamflow dynamics.

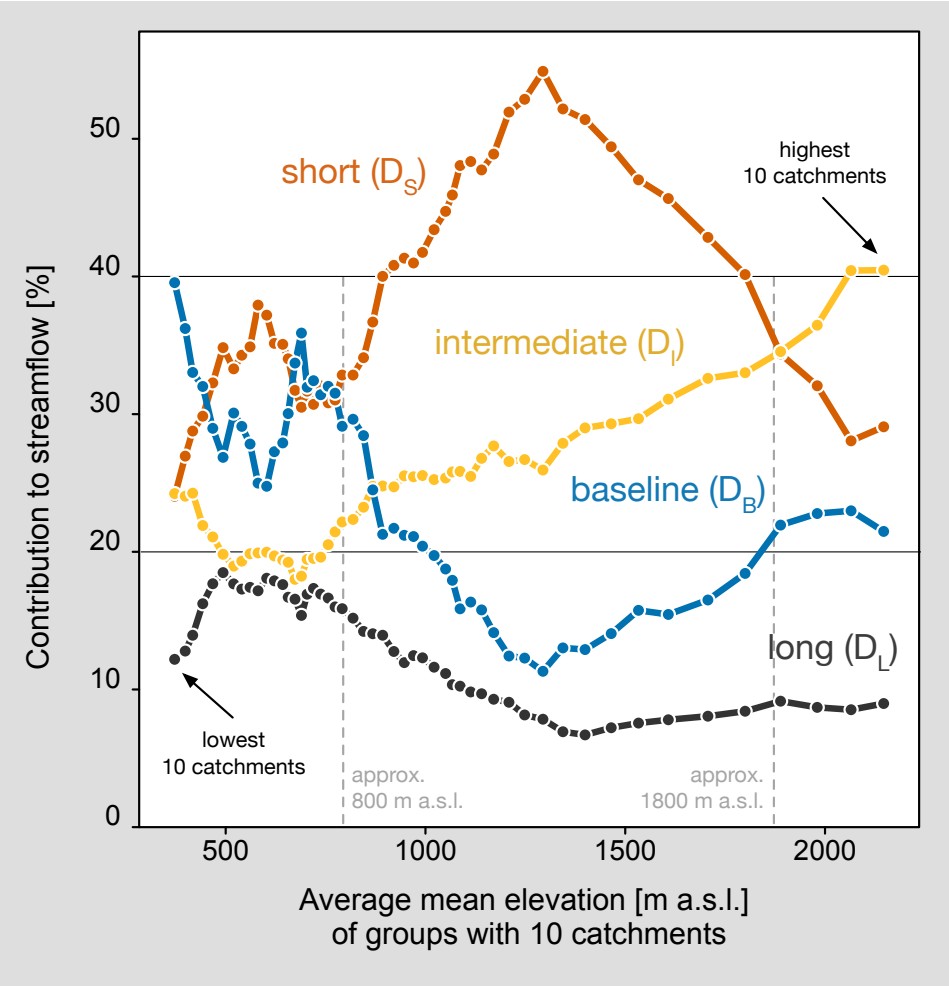

**Figure 7: Relationship between elevation and contributions to streamflow in different delay classes. Catchments are sorted according to mean elevation and grouped into sets of 10 catchments to estimate average mean elevation for each group.**





## 4.4 Colwell's Predictability for attribution of delayed contributions

Following Colwell's Predictability ($P_T$) measure, streamflow predictability is composed of constancy ($P_C$) and seasonality ($P_S$). Adding up $P_C$ and $P_S$ reveals a distinct *U-shape* pattern for $P_T$ (Fig. 8a). Overall, $P_T$ of RLWR and SNOW catchments is higher than for RUPR and HYBR catchments. The lower $P_T$ provide insights into the catchment characteristics of HYBR, and partly RUPR catchments, as smaller contributions in $D_L$ and $D_B$ identify catchments with smaller storages. As HYBR catchments are mainly controled by $D_S$ contributions we attribute a smaller storage capacity and less water rentention potential to those catchments. A higher $P_T$ is mainly attributable to higher $P_C$ in RLWR and higher $P_S$ in SNOW catchments. Including a correlation analysis (Fig 8c) we found strong relationships between $D_B$ contributions and $P_C$ (r = 0.61) and between $D_I$ contributions and $P_S$ (r = 0.82). Interestingly, $P_C$ for SNOW catchments is not markedly lower compared to the other three catchment groups. Higher $P_C$ is an indication of higher streamflow sustainability throughout the year and this sustainability is related to $D_B$ contributions. Correlation analysis (Fig. 8c) also reveals that $D_S$ contributions is negatively correlated to $P_T$ (r = -0.47) and $P_C$ (r = -0.45).

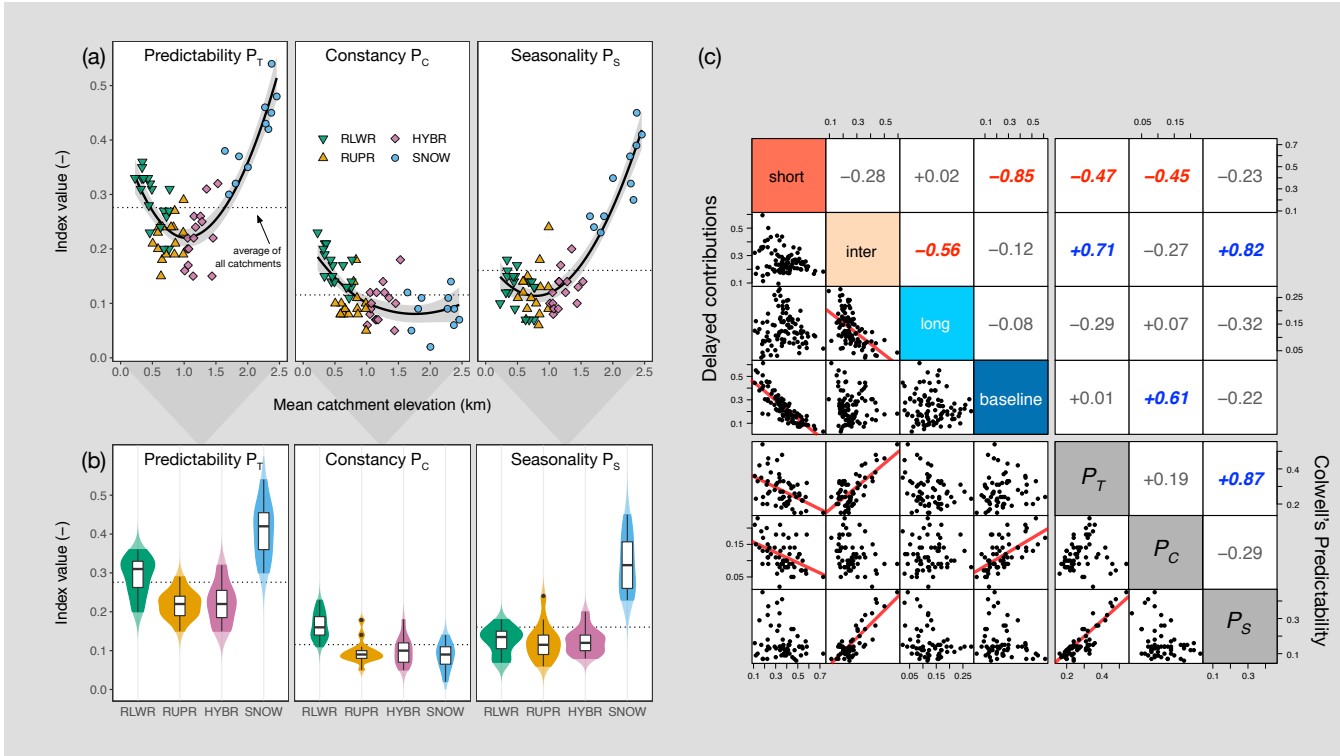

**Figure 8: Overview of Colwell's Predictability ($P_T$) = Constancy ($P_C$) + Seasonality ($P_S$) for (a) all catchments and (b) the four catchment groups (higher saturation of violin plots delimits the data range). (c) Relationship (i.e. Pearson's correlation coefficient) between relative streamflow contributions in the four delay classes and the components of Colwell's Predictability ($P_T = P_C + P_S$). Coloured coefficients are statistically significant (*p-value* < 0.001).**



## 5. Discussion

### 5.1 Technical aspects of delayed flow separation

Any discussion of the applicability of the UK-IH baseflow separation method has to account for the fact that the methodology

was developed for humid- and rainfall-dominated catchments (Gustard et al., 1992) and the conventional block size $N = 5$ is not necessarily valid for catchments with a different climate and hydrological regime, such as lake- or snow-dominated catchments (WMO, 2009). It provides a first-order estimate of catchment responsiveness, separating the streamflow into a fast and a slow component, and has proved useful in many studies around the world. For some studies' objectives, however, the BFI and the two-component baseflow separation into binary quick- and baseflow has too strong limitations due to e.g. arbitrary

separation parameters or the mixture of different delayed sources into one baseflow component (Hellwig and Stahl, 2018; e.g. Kronholm and Capel, 2015; Parry et al., 2016; Partington et al., 2012). In this study, we introduce a new index, the Delayed Flow Index (DFI), which allows a range of different delayed sources to be assessed an may hence provide an alternative for more complex systems. Comparing DFI and BFI we found a relatively consistent ranking between BFI and $DFI_{60}$ values with a Spearman's rank coefficient $p = 0.83$. However, the $DFI_{60}$ values varied between 20 and 80% of the corresponding BFI

values among the catchments included in our study. Accordingly, we argue that $DFI_{60}$ may provide a better measure of the catchment in terms of its ability to sustain low flows for sustained dry periods. This is important knowledge for assessing the resilience of aquatic ecosystems, improve water resources management (e.g. for quantification of residual water amount in streams) or test low flow sensitivity to climate change (e.g. change of $DFI_{60}$ over time).

The wide variety in the shape of CDCs can be seen as reflecting the wide range of catchments spanning from erratic (more

flashy) to persistent (more stable) streamflow regimes (Botter, 2014). Accordingly, we identified catchments with shorter and more intermediate delayed contributions as compared to more long and baseline delayed contributions to streamflow. The fraction of the flow contributions within each delay class is, however, depending on the number of delay classes, i.e. the number of breakpoints and $N_{max}$. In particular RLWR catchments showed a relatively wide range of breakpoint estimates. Both fewer or more breakpoints are feasible to imbed as long as the breakpoints represent the stepwise decrease of the slope and the

shape of the CDC (Miller et al., 2015). Also, an adjusted value of $N_{max}$ might be needed for other climates or streamflow regimes.

The decreasing CDC slope and the piecewise linearity of CDCs is a meaningful tool for hydrological analysis as breakpoints might identify a specific point in time during receding streamflow when a faster source stops to contribute and slower contributions then maintain flow and control streamflow dynamics. Nevertheless, a definite attribution of delayed streamflow

contributions to specific sources within a catchment is technically only feasible if a catchment's fingerprint of contributing sources is known and underlying processes are understood. Hence, the DFI is separating different components of the streamflow hydrograph based on their delay patterns and not based on their source identification. A clear classification of delayed streamflow contribution seems to be more challenging if streamflow is less consistently controlled either by rain or snow. Our findings of a shift in the catchment response from rainfall- to snow-dominated (at around 1800 m a.s.l., ref. Fig. 6),





for example, support the results of a soil moisture analysis in Switzerland suggesting that around 2000 m a.s.l., the regime controls change from precipitation/evaporation to more frost-affected (Pellet and Hauck, 2017). In this respect, another potential future application of DFI may be the separation of snowmelt and glaciermelt contributions to streamflow with an additional breakpoint. Some of our study catchments are partly glacierized (<7%) and glacier melt in headwaters during warm and dry summers will eventually make a significant streamflow contribution. However, due to the small proportion of

glacierized catchments in our data set, we did not separate between snow- and glaciermelt in catchments represented both by the intermediate delayed component. One useful future approach might be to investigate CDCs of years with more/less snowmelt and more/less glaciermelt (i.e., in total four combinations) to identify the specific delays of those contributions or alternatively, to perform a seasonal analyse of DFI values, e.g. during specific "melt months" (i.e. May versus August in Switzerland). However, the scope of this paper was not to identify the specific sources in each catchment, but rather to present

a framework for attributing patterns of delayed contributions to (potential) dominant sources including rainfall, snowmelt and groundwater. From a practical perspective the proposed method is data parsimonious and has a high potential for hydrological application worldwide (due to readily available streamflow data) and can be also adapted for other regimes e.g. for intermitted streams with zero flows in semi-arid regions as suggested by Aksoy et al. (2008) or other variables (e.g. precipitation, groundwater).

The decision to use the smoothed minima method instead of the also well-established recursive filter procedures (Eckhardt, 2008; Nathan and McMahon, 1990; Smakhtin, 2001) had the advantage that the choice of block size $N$ (in days) can be directly related to catchment response and thus, usable for interpretation of main sources of streamflow and as such generally more accessible compared to parameterization of recursive filters (e.g. recession parameters) and their forward- and backward-filtering procedures. However, a preliminary analysis showed that using a common recursive filter procedure (Nathan and

McMahon, 1990) lead to the same ranking of BFI ($DFI_5$) values as the original IH-UK-method (results not shown). Similarly, it would be possible to systematically vary parameters in recursive filter procedures to derive different BFI values for different recession parameters, which may represent a faster and slower recession behavior of catchments (i.e. short- and long-term recession behavior).

**5.2 Paradigm shift from quick- and baseflow to delayed flow**

Splitting contributions into two main categories, i.e. fast and slow, has proved to be useful as a simple measure of catchment responsiveness. Several studies using hydrograph separation with empirical parameter values, e.g. fixed block size N or fixed recursive filter parameter like suggested by Nathan and McMahon (1990), ignore that different environments have a different type and number of storages and hence, various delayed contributions to streamflow, which also may be highly depending on

season. Even if, for example, a snowmelt pulse is considered as a baseflow contribution to streamflow, the higher BFI value should not be attributed to (large) groundwater storages, but instead to the snowpack and snowmelt processes and their seasonality. Furthermore, especially in large sample hydrology single specific catchment features like the proportion of lakes,





wetlands or reservoirs are not considered appropriately in two-component hydrograph separation as often climate variability is used to explain streamflow variability. With DFI analysis two catchments with the same climate will have different CDCs

if e.g. one has streamflow contributions from a lake or reservoir and the other has not. In this respect, delayed flow contributions can be seen as "response patterns" and accompanied recent efforts to focus more on effect tracking instead of particle tracking to understand streamflow components and streamflow generation processes (Weiler et al., 2017). Breakpoint estimates are particularly helpful to support this effort as their positions on the CDCs can be interpreted as the maximum delay of a faster source ($N = BP$). Beyond the breakpoint, the streamflow contribution of the faster source diminishes and streamflow variability

is more and more controlled by the slower source ($N > BP$). According to the breakpoint analysis, quickflow can be considered as the short delayed component ($D_S$) that ceases to contribute after 2, 3 and 4 days in 55%, 70% and 95% of all our study catchments, respectively. Comparing $D_S$ contributions (DFI) with quickflow contributions (BFI; $N = 5$) we found on average 11.5, 7.0 and 2.9 % less short delayed contributions to streamflow for catchments with their first breakpoint $BP_1$ at 2, 3 or 4 days, respectively. Differences between BFI-quickflow and $D_S$ contributions are higher for HYBR and RLWR catchments and

less pronounced for SNOW and RLWR catchments. It is important to note that the position of a breakpoint depends on the total number of breakpoints on the CDC and on the value of $N_{max}$.

Based on the breakpoint and DFI analysis, we follow the recommendation of Hellwig and Stahl (2018) to integrate catchment-specific response times in low flow analysis and hydrological modelling. The authors show that it is impossible to distinguish the contributions from groundwater and snowmelt in snowmelt-dominated catchments with a two-component baseflow

separation. We found from CDCs that the slowest dynamic contributions have response times with mean values between 28 and 45 days in the long delay class ($D_L$). The governing time scales of streamflow dynamics are also subject of other studies. Brutsaert (2008) reviewed storage coefficients in comparison with recession analysis and identified characteristic drainage processes on timescales of $45 \pm 15$ days, but his study catchments had a larger average catchment area compared to this study. Staudinger and Seibert (2014) estimated streamflow persistence in various pre-alpine and alpine catchments (3-350 km$^2$) and

found even in quickly responding catchments with assumingly small storages, the slowest delayed signal to be around 50 days. Compared to these results, our estimates of $D_L$-contributions are rather small, showing on average delayed streamflow contributions over around 42 days. Andermann et al. (2012) found the time lag between precipitation and discharge with hysteresis loops in high-alpine catchments in central Himalaya to have a characteristic response time of 45 days. In a pan-European modelling study, Longobardi and Van Loon (2018) separated response patterns of catchments into poorly-drained

(BFI < 0.5) and well-drained (BFI > 0.5) catchments and assigned estimated delay times of the slowest storage in the model to be $48 \pm 14$ days and $126 \pm 47$, respectively. This gives some evidence that for mostly groundwater-dominated catchments, $N_{max}$ may be set to a larger value (here 60 days) to better capture small variations in year-to-year low flow magnitude. However, also smaller delays around 60 days for groundwater-dominated systems were found (Huang et al., 2012). A 60-day 'seasonal' period has also been reported as an appropriate window size to characterize the variability of streamflow regimes in respect to

environmental flow and ecohydrology (Lytle and Poff, 2004; Poff, 1996). Accordingly, our results suggest that the DFI method (with $N_{max}$ set to 60 days) is able to capture the time scales of the most relevant dynamic drainage processes and response





patterns between 1 and 60 days. Our findings are consistent with studies revealing that a high proportion of streamflow is less than three months old (Jasechko et al., 2016). However, we recommend further evaluating of the methodology, in particular the breakpoint-delayed flow-separation in catchments where contributing sources to streamflow are well understood and

timescales of contribution are already estimated by isotopic or solute measurements with, for example, end-member mixing analysis (Miller et al., 2016).

### 5.3 Attribution of delayed flows to catchments' storages

In lowland catchments, recharge is crucial to maintain a constant year-to-year groundwater contribution to streamflow, in alpine catchments the seasonal snowpack controls a large part of streamflow contributions and maintains streamflow during

the summer period (e.g. due to meltwater or saturated soils that allow for groundwater recharge). It has been suggested that the latter contribution will gradually decrease in the future due to global warming, related shift of snowmelt season and elevation-dependent warming (Pepin et al., 2015). However, projections of future groundwater recharge in lowland catchments are complex due to the combined effects of changes in precipitation, evapotranspiration, land use and land cover as well as in human water demand. The analysis of delayed contributions (Fig. 7) reveals that for our study catchments, certain elevational

thresholds can be established. For example, $D_S$ contributions show highest relevance at around 1300 m a.s.l., but then decrease for higher elevations. Above this elevation water stored in snow has increasing influence on streamflow contributions (higher $D_I$ contributions), but also $D_B$ contributions (e.g. groundwater) increase and play an important role in high elevation catchments. Regarding the smaller subsurface catchment storages for HYBR and partly RUPR catchments (i.e. lower $D_B$ contributions), the results suggest that low predictability of streamflow (i.e. Colwell's Predictability $P_T$) is caused by a high

amount (up to 60%) of $D_S$-contributions. The streamflow variability in these catchments is highly sensitive to rainfall, evapotranspiration and fast runoff processes throughout most of the year (Fig. 8). But, if geology rather than climate was a first-order control of groundwater storage, high alpine geology would exhibit functional units that provide slowly draining groundwater stores in extended dry seasons.

Parry et al. (2016) have shown that elevation outperforms BFI as a measure to characterize the spatial variability of catchments'

responsiveness in the UK. Hence, BFI might not be sufficient to capture the dominant delayed contributions to streamflow across different streamflow regimes. Whereas for some regions, such as the UK, a linear relationship between elevation and responsiveness will be sufficient, the U-shape of Colwell's Predictability (Fig. 8a) suggests that higher streamflow predictability in our study region can be caused by different delayed contributions (i.e. $D_I$ and $D_B$ contributions). This justifies using multiple delayed components during response analysis. For example, we found that a 60-day block size captures virtually

all the variability in the annual minimum flows of the catchments (Fig. 2 and 5). Minimum annual flow is sustained by a rather constant delayed contribution with slower and deeper pathways with minimal variations from year to year. The baseline contribution ($D_B$) has a smooth seasonal variability and accounts for up to 60% of mean streamflow in our study catchments (mean 25%). Focusing on the range between the 5th and 95th percentile of all 60-day minimum flows, the contributing storages contribute to the flow with annual averages between 70 and 350 mm. This amount is equivalent to 6-24% of annual rainfall





and 11-53% of annual streamflow. Interestingly, this ratio is not much smaller in alpine catchments (catchment group SNOW), where $D_B$ accounts for 12-24% of annual precipitation, and 15-39% of annual discharge. Smaller absolute $D_B$ contributions with increasing elevation could be expected due to less developed soils, steeper slopes, and more direct topography-driven runoff generation. Interestingly, for the whole year we found higher or equal relative $D_B$-contributions for SNOW catchments (22%) compared to RUPR (21%) and HYBR (14%) catchments, and the highest absolute $D_B$ contributions for SNOW catchments of 250 mm/a compared to around 150 mm/a for the other catchment groups (Fig 5b, Fig. 6). However, during summer season RLWR and RUPR catchments have larger $D_B$ contributions compared to HYBR and SNOW catchments due to the dominance of $D_I$ contributions (i.e. snowmelt) in SNOW and $D_S$ contributions in HYBR catchments or longer periods in summer without rain in lower-elevation catchments (i.e. less $D_S$ and $D_I$ contributions in RLWR and RUPR).

Nevertheless, $D_B$ contributions in SNOW catchments during winter are considerably high and these findings contradict the assumption of small or negligible subsurface water stores in high-elevation catchments. Along with other recent studies (Staudinger et al., 2017), we found compelling evidence (cf. higher level of CDCs for SNOW catchments during winter season in Fig. 5b) to reconsider the hydrological role of storages beyond snow storage in alpine environments. According to our CDC analysis winterly recession in high-elevation catchments (SNOW) are likely the results of slowly draining subsurface water stores. This is underpinned by the "frozen state" that the SNOW catchments have during winter; precipitation is stored in the snowpack, snowmelt is not occurring and recharge impulses are comparably low. Thus, subsurface storages (e.g. groundwater) are responsible for sustaining flow during winter. Weekes et al. (2014) argued that depositional, often paraglacial landforms with colluvial channel, talus and rock glacier features are good indicators of higher recession constants and thus high water storages indicating slower draining and more sustained baseflow. Amongst others (Clow et al., 2003; Hood and Hayashi, 2015; Miller et al., 2014; Roy and Hayashi, 2009; Staudinger and Seibert, 2014), our analysis suggests that beside transient snowpack storage also diverse groundwater storage units in alpine catchments (e.g. glacier forefields, taluses, gravel banks, and other colluvial features) are likely important subsurface storages sustaining streamflow and downstream water availability. Water drainage from these units is especially important during the snow accumulation period in winter and subsequently in years with reduced snowmelt in summer. Talus fields, for example, can contribute more than 40% to streamflow and sustained baseflow after the snowmelt period (Liu et al., 2004). Estimates of total storage volume and a comprehensive understanding of the recharge cycle of those storage units are missing so far, but Paznekas and Hayashi (2015) assumed, for multiple alpine catchments in the Canadian Rockies, that groundwater storage is completely filled up every year and described alpine groundwater as an important streamflow contribution. Hood and Hayashi (2015) estimated peak groundwater storage amount (60-100 mm a$^{-1}$) in a small headwater located above 2000 m to be roughly 5-8% of mean annual precipitation and 9-20% of pre-melt snow water equivalent. Also in semiarid mountainous regions, groundwater is supposed to be a major streamflow contribution, sustaining water availability downstream (Jódar et al., 2017). In accordance with the above studies, we found that high-elevation catchments have larger catchment storages than previously thought and thus, the results of our data-driven





analysis agrees well with methodological more advanced studies in the same (Staudinger et al., 2017) or similar regions (Hood and Hayashi, 2015).

## 6. Application of DFI: Seasonal sensitivity of low streamflow based on delayed contributions

In this section a potential application of the DFI is given for ten randomly selected catchments of each catchment group. The low flow stability ($S_{LF}$) index compares the average streamflow of the four driest months of a given month to the average flow of that month (Sect. 2.6). Higher $S_{LF}$ values indicate higher flow stability during dry months. The selected SNOW and RLWR catchments have an average $S_{LF}$ of 45-50%, whereas the selected HYBR and RUPR catchments show smaller $S_{LF}$ between 20-35% (represented by the length of the colored bars in Fig. 9). $S_{LF}$ can be combined with information on contributing sources

for different months and streamflow regimes as highlighted by the stacked coloring of the bars (note that each color stack is scaled to the length of the total bar and thus not comparable in size across catchments and months). Dashed boxes in Figure 9 show the month with the lowest monthly streamflow for each of the four catchment groups. SNOW (winter low flows in January) and RLWR (summer low flows in August) catchments have around 75% to 80% $D_L$ and $D_B$ contributions during low flow season. Compared to that HYBR (winter low flows in January) and RUPR (summer low flows in August) catchments

have only around 60% $D_L$ and $D_B$ contributions during low flows. Combining information about low flow stability and streamflow contributions with different delays offers new insights in the drivers of seasonal low flows across different catchment regimes. Extending the analysis on more months (March and October in Fig. 9), in March SNOW catchments have still in average 80% $D_B$ contributions, whereas the other catchment groups have around 50-60%. Here the analysis points to the likely role of potentially slowly draining groundwater aquifers that must be recharged to sustain streamflow during late

winter and early spring before the snowmelt pulse, i.e typically in April or May (Table 1). In August and October, the picture is changing, high-elevation catchments (SNOW) have then in average large $D_I$ contributions (45% and 30%) compared to January and March (15% and 5%) whereas low-elevation catchments (RLWR) have large $D_B$ contributions during dry months in summer (65% in August and 80% in October).

These contribution patterns are in accordance with Zappa and Kan (2007) who found that catchment elevation can explain

more than 80% of the hydrological response during the Swiss 2003 heatwave. In 2003 the snowmelt buffered low flows in higher elevation catchments and lowest flows (i.e. ratio to long-term average streamflow) were found below 1000 m a.s.l. where subsurface water stores (e.g. groundwater) were no longer able to sustain streamflow. HYBR catchments are in general more sensitive to precipitation deficits during dry months in summer due to lower $D_B$ contributions (20% in August, 30% in October) and higher $D_S$ contributions (50% in August, 45% in October) (Fig. 9). Focusing on summer (low) flows the analyses

suggests that decreasing snowpacks (SNOW), decreasing summer precipitation (HYBR) or changes in recharge mechanisms (RLWR) might lead to reduced streamflow and to eventually new or intensified streamflow droughts. Incidentally, changes in headwater snowpacks can also have severe consequences on groundwater recharge in lowland basins (Ameli et al., 2018). As





the snowmelt season in high-elevation catchments is also relevant for the downstream freshwater availability, low flow stability

and DFI analyses can give important insights in the water availability for large river basins (e.g. Rhine basin).


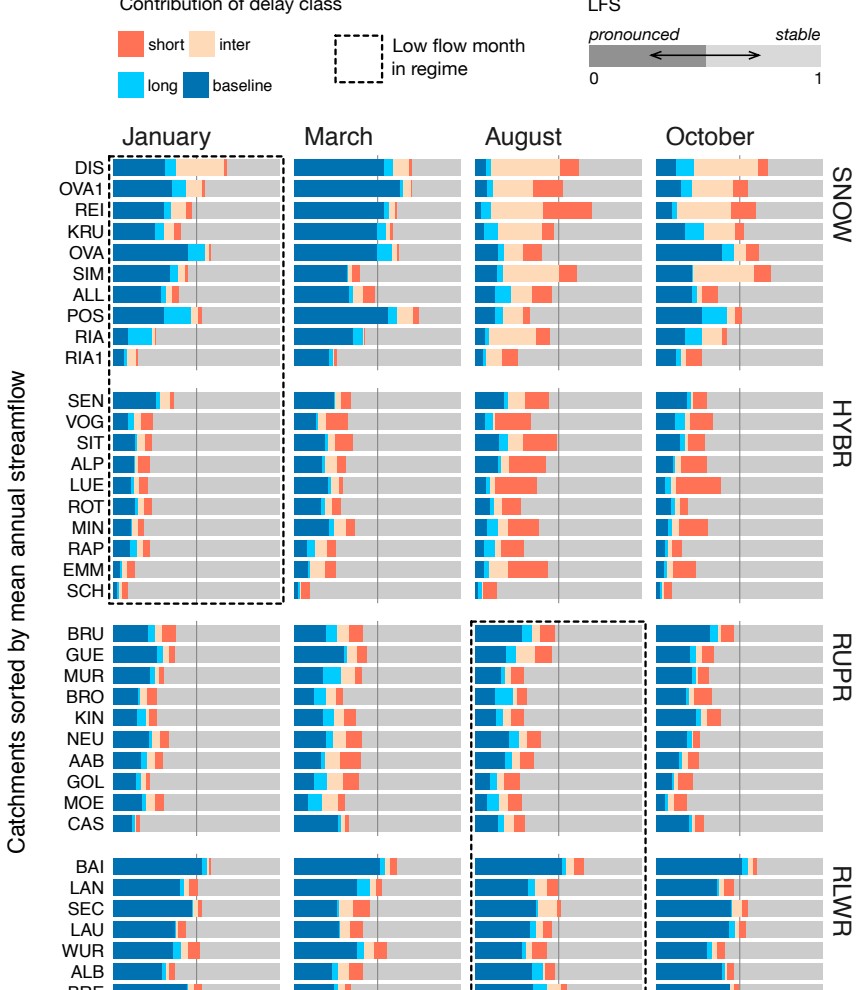

**Figure 9: Low flow stability index ($S_{LF}$) for four different months shown for the four catchment groups, each composed of ten randomly selected catchments. The month with lowest monthly streamflow across the catchment groups is marked with dashed boxes. Stacked colouring of the bars represents the composition of different delayed contributions to streamflow (scaled to the total**
**length, which is here equal to the value of $S_{LF}$).**



## 7. Conclusions

We advanced a commonly used binary quickflow-baseflow separation method and introduced a novel concept of delayed flow index (DFI) based on short, intermediate, long and baseline delayed contributions. Testing the DFI for a set of 60 mesoscale catchments revealed that catchments along a pronounced elevational gradient have characteristic delay curves and sets of unique breakpoints. The breakpoints in these curves identify different streamflow contributions with different controls on streamflow regime. Our analysis shows that for headwater catchments in Switzerland and south-west Germany covering a pronounced elevation gradient, short delayed contributions (i.e. quickflow) cease 2-4 days after hydrograph peaking and baseline delayed contributions (delays with > 60 days) control the magnitude of streamflow sustainability. The continuous analysis for delays between 1 and 60 days is one of the major differences compared to two-component BFI analysis with two delayed components (delay smaller or larger than 5 days). The response-oriented perspective on streamflow contributions supports a more comprehensive analysis of different catchment storages revealing that groundwater and snowmelt are often mixed in one baseflow component in binary baseflow separation given that the whole year is considered. In addition, intermediate delayed contributions can have a strong influence on the streamflow regime. Hence, the proposed DFI allows a more physically meaningful insight to governing processes than the two-component separation procedures, and thus represent a step towards an attribution of delayed contribution to potential sources (stores). The distribution of different delays across catchments improves our understanding of catchment storage across streamflow regimes, drivers of low flow variability in different seasons, and allows quantifying streamflow sustainability.

The notable high baseline delayed contributions even in alpine catchments further support the need to reconsider the role of alpine groundwater storages, which may indeed be larger than previously thought (Staudinger et al., 2017) and hence, to reject the former concept of Teflon catchments (Williams et al., 2016) in these environments. Baseline delayed contributions in high elevation catchments can account for around 25% of annual precipitation and 40% of annual streamflow. Study catchments in between approx. 800 and 1800 m a.s.l. show the highest low flow sensitivity to climate variability due to smallest catchment storages and high amount of short delayed contributions to streamflow. The DFI analysis suggests that the severity of low streamflow in summer is controlled by snowmelt (in higher elevation catchments) and groundwater or other delayed sources (in lower elevation catchments) advocating that changes in corresponding recharge mechanism and subsurface storages could lead potentially to more severe low flows in the future.

## Code availability

The R Code to calculate the Delayed Flow Index is available in the R package *lfstat* (Koffler et al., 2016). Within the package, the function *baseflow* allows calculating delayed flow time series based on the parameter *block.len* (N). Breakpoints of CDCs can be calculated with R package *segmented* (Muggeo et al., 2008).



**Data availability**

Data is not freely available, but streamflow data can be accessed through the agencies.

**Author contribution**

MS, TS, MW and LMT had the research idea and designed the study. MS performed the analyses and developed the delayed
flow index framework together with TS and MW. All co-authors edited the manuscript.

**Competing interests**

The authors declare that they have no conflict of interest.

**Acknowledgments**

Streamflow and catchment metadata were provided by the Environment Agency of the German state of Baden-Württemberg
(LUBW) and the Federal Office for the Environment (FOEN), section of hydrology in Switzerland (BAFU). The authors thank
Maria Staudinger and Benedikt Heudorfer for their comments during manuscript drafting. The study is also a contribution to
UNESCO IHP-VII and the Euro-FRIEND project.

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
