# Peer review of "Beyond binary baseflow separation: a delayed flow index for multiple streamflow contributions"

_Hydrology and Earth System Sciences, 2019_

## Referee Comment (RC1) · Anonymous Referee #1 · 16 Jul 2019

This is an interesting article and essay about an approach that may find its place in practice. It aims to subdivide total (merged) baseflow (slow flow) into its possibly different components. The title, however, appears slightly high-handed. This enhanced application of the smoothed minima method will hardly replace other baseflow separation methods; hence it is not "beyond" but may be "besides". The splitting of flow contributions is a fresh idea, but not a new one. Also, the term "delayed flow" appears problematic. A delay is normally a time shift which cannot really describe the inflow-outflow (retention) processes of reservoirs (aquifer, snow, lakes...). The paper is not easy to read. Many formulations could be straighter forward. Lines 66-77: This section gives the impression that hydraulic processes are not fully understood. Aquifers act as

reservoirs discharging baseflow according to hydraulic head (pressure) and rather not "water that is moving slowly..." (line 70). Skip or rewrite. Also, the many abbreviations (e.g. in chapter4.1) and awkward formulations in chapter 5 make reading difficult. The proposed method is built on the IH-UK smoothed minima method following the philosophy of former respective research work performed in Freiburg at the institute of three of the authors under the denomination Wundt/Kille-Demuth method (Demuth 1989). It is an empirical, statistical approach to only detect and describe the effects of storage in aquifers etc. on streamflow and does not model the hydraulic processes. Baseflow separation methods based on reservoir algorithms are not even mentioned in the present paper though they are the closest to physics and hydraulics. Line 78 and others: Write Hollick instead of Hollwick. Abstract and other places: The authors criticize contemporary "binary" baseflow separation methods "for their arbitrary choice of separation parameters". This is not quite an objective argument. So, like "the DFI is based on characteristic delay curves...", other baseflow separation models are calibrated with observed flow recessions and yield good results. The authors probably used data of a number of the same stations as their colleagues in Bern, Switzerland (Meyer et al.2011), who report: "Three different procedures to separate baseflow are applied in 59 catchments in Switzerland. The results show a good coherence of baseflow with well-known storage processes". Why not have a look? The authors criticize baseflow separation methods because they "merge different delayed components". Reservoir based separation methods were applied for distinguishing and quantifying different contributions, two examples: Schwarze et al. (1989) created a model of parallel linear reservoirs representing different contributing aquifers or storages. Wittenberg (2003) distinguishes with his nonlinear reservoir method groundwater outflow, groundwater evapotranspiration, abstraction.... Is the present method only or particularly suited for regional studies since a linking of flow contributions to catchment characteristics is needed? Line 485: Groundwater recharge does not need saturated soils. Infiltrating water passes the vadose zone via preferential pathways.

Demuth, S. The application of the West German IHP recommendations for the analysis of data from small research basins. Friends in Hydrology, IAHS Publication No. 187, 43-60, 1989. Meyer, R., Schädler, B., Viviroli, D., Weingartner, R. Die Rolle des Basisabflusses bei der Modellierung von Niedrigwasserprozessen in Klimaimpaktstudien. (The role of baseflow in modelling low-flow processes in climate impact studies). HyWa 55,5,244-257, 2011. Peters, E. van Lanen, H.A.J. Separation of base flow from streamflow using groundwater levels – illustrated for the Pang catchment (UK). HProc 5548, 2003. Schwarze, R., Grünewald, U., Becker, A., Fröhlich, W. Computer aided analysis of flow recessions and coupled water balance investigations. Friends in Hydrology, IAHS Publication No. 187, 75-83, 1989. Wittenberg, H. Effects of season and man-made changes on baseflow and flow recession: case studies. HProc 17, 2113-2123, 2003.
* * *

---

## Referee Comment (RC2) · Anonymous Referee #2 · 19 Jul 2019

The manuscript presents an interesting idea to distinguish different baseflow components. To my understanding, the main methodological assumptions are correct and could potentially make an interesting contribution for the journal. However, in my opinion, the draft is not well structured and written making it difficult to read, uncertainties regarding the selected dataset and the separation of regimes need to be addressed and the discussion and conclusion should be revised accordingly prior to publication. Below there are some suggestions that might help to improve the manuscript:

Major comments: - The overall readability should be improved by favoring short and straight forward formulation: The use of a multitude of abbreviations and the inconsis-

tent usage of wording (e.g. with regard to the term storage) make the paper tough to read: e.g. the introduction needs to be re-written, in my opinion it lacks structure and conciseness; there are many incomprehensive formulations and inconsistencies e.g. the sentence in line 30 to 31 does not make sense to me, are you talking about the magnitude of "sustained streamflow" or more generally of the existence of streamflow? "sustained streamflow and hence freshwater availability" – most freshwater is stored in aquifers; And why does streamflow need to be estimated from BFI? This first general introduction is just very confusing. line 34: What are stored sources? clearly, discharge is coming from "stored sources" whenever it is not raining, the BFI is often interpreted as the contribution from groundwater ... as you state in line 38.Lin e 39: you write about water from groundwater, soil and "other delayed source" Which other sources do you mean? Please mention them! There are multiple more examples in the following lines, please try to be more concise in your wording and restructure the introduction!

- The way you report the selection of catchments is critical: you state that human influence on these "headwater" catchments is negligible; (line 254-255): the term headwater catchments is a little misleading for basins of up to 955km2; most of the area in Germany and Switzerland is densely populated, thus human influence might matter: especially when overall magnitudes are small e.g. distinguishing between long delay and baseline delay you will need to make clear that we are not looking at human influence or potential feedbacks from evapotranspiration and vegetation during extended dry periods. The MAG is a regulated basin with huge dams for hydropower and thus highly damped discharge, which makes me a little suspicious if the other selected catchments are suitable for the analysis; please remove MAG and consider double-checking your catchment selection!

- the reasoning for classifying the different regimes, especially when distinguishing between RLWR and RUPR, needs to be further discussed: looking at figures 5 and 6 one could argue that the variation within the groups RLWR and RUPR is larger than the difference between their medians, so from a process point of view (in the end

that's what you want to capture) the separation based on mean and max elevation might not be suitable. In Figure 7 you even argue that different elevation classes might be more representative. HYBR represents a mixture of snow and rainfall dominated catchments, but obviously as suggest by your results, it is not, can you discuss why? There might not be an easy solution to these issues, but maybe they can be discussed more detailed. The snow-dominated catchments are significantly smaller than all other catchments (Table 2), please mention that explicitly and update your interpretation accordingly (e.g. line 294 "higher flashiness during summer flows" might be an artefact of catchment size); maybe you can provide some basic streamflow statistics of the dataset e.g. in Table 1 potentially add magnitude and variation of q5, q50 or recession characteristics with respect to the selected catchment grouping

- also the discussion would benefit from restructuring and improved consistency: e.g. line 409 where can I see "a shift in catchment response" at around 2000m? line 420: how would you apply the framework worldwide? your case study is on data carrying a strong seasonal signal and elevation gradient In my opinion the called paradigm shift appears a little too ambitious, as there are (as you also point out in the introduction) several approaches to capture delayed contributions from different storage settings. I don't see how the proposed approach assess (line 439) "different type and number of storages, hence various delayed contributions" While I agree that BFI does not account for single catchment features, also DFI will not identify them specifically (line 445), but you rather get a signal of delayed outflow from potentially multiple (different) sources. Potentially the climate regime itself might significantly influence Nmax, dry periods in southern Europe or norther latitudes, high-elevation catchments streamflow droughts occur on timescales of < 60 days (up to 4 months). Whereas it might not be relevant for large parts of your study region, it might lead to a biased view on snow-dominated systems and potentially when applying the proposed method elsewhere. Also 5.3 starts with a confusing argumentation (lines 483, 484): recharge is crucial everywhere, fair enough, in Alpine catchment seasonal snowpack supplies summer streamflow, however according to table 1 low flow / delayed flow occurs Jan to March, also (line 485)

saturated soils are not allowing groundwater recharge. The influence of global warming on melt processes and groundwater recharge is highly depended on the elevation range you are referring to (line 486) To my knowledge it is not yet clear if smaller DB (or smaller groundwater contributions in general) can be directly related to the size of subsurface storages (line 493). There is ongoing discussion if differences in magnitudes are related to variable connectivity of storage and stream, variable precipitation / evapotranspiration in different elevation / exposition or differences in storage recharge. Line 513: If DB is the groundwater contribution, why would less developed soils matter? Again, the ranges you report a quite large, however the SNOW catchments are significantly smaller. The whole argument on storage in SNOW catchments is complicated to follow, you start the argument with Alpine storages are small (but you don't mention who reports that), afterwards you mention numerous studies that report the opposite, to conclude that "high-elevation catchments have larger catchment storage than previously thought". Some final thoughts on 6: Low streamflow occurrence might be highly variable comparing different years, mainly depending on climate, I'd suggest mentioning that explicitly and re-formulate less definite. Also the high accordance to elevation gradients might be specific for the Alps, you might not find that in other regions e.g. Scandinavia, southern US;

Minor comments: in Figures 1, 3 & 9 the difference between light blue and blue (long vs. baseline) is not visible (in Figure 7 you even replace blue by black, which makes it much more readable, maybe change it also for Figures 1 and 3)

the usage of hyphens is quite arbitrary throughout the document, to my knowledge there are clear rules, please check them and change accordingly e.g. line 26 low flow stability index… low flow regimes, line 30 groundwater-surface-water-interactions, line 318 5-days,… In Figure 1 the dark blue color refers to baseline delay class although it is obviously (the volume) below the baseline, 1b is too small

In line 169: What is the "seasonal low flow period"? How long is it? Where can I see that period of 60 days in the hydrographs of Figure 1a? What exactly is AM, MAM and

MQ and how do you calculate them?

Line 387: assessed, and may; line 391: sustain low flow for sustained dry periods; line 523: winterly recession;

---

## Author Comment (AC1) · 9 Aug 2019

**We would like to thank the reviewer for the feedback and the suggestions to improve the manuscript. Here, we respond to each comment (in bold).**

This is an interesting article and essay about an approach that may find its place in practice. It aims to subdivide total (merged) baseflow (slow flow) into its possibly different components. **Thanks for this general comment.**

The title, however, appears slightly high-handed. This enhanced application of the smoothed minima method will hardly replace other baseflow separation methods; hence it is not "beyond" but may be "besides". The splitting of flow contributions is a fresh idea, but not a new one. Also, the term "delayed flow" appears problematic. A delay is normally a time shift which cannot really describe the inflow-outflow (retention) processes of reservoirs (aquifer, snow, lakes. . .).

**We do not intend to replace any baseflow separation method. We argue that going beyond binary baseflow separation might be valuable for certain applications (e.g. in catchments with more than two dominant streamflow contributions or in highly seasonal regimes).**
**We suggest a new, more condensed title for the paper: Beyond binary baseflow separation: a delayed flow index for multiple streamflow contributions.**
**The term "delay" is used as a more generic term (not related to a particular process) to describe the response patterns of different contributions (aquifer, snow, lakes) on the dynamic of the hydrograph: a short-delayed contribution is water that is moving quickly through the hydrological system controlling the peaks of the hydrograph, a long-delayed contribution is moving more slowly controlling the tailing and recession behavior of the hydrograph). Schwarze et. al (1989) have also used the term "delayed long-term base flow".**

The paper is not easy to read. Many formulations could be straighter forward. Lines 66-77: This section gives the impression that hydraulic processes are not fully understood. Aquifers act as reservoirs discharging baseflow according to hydraulic head (pressure) and rather not "water that is moving slowly. . ." (line 70). Skip or rewrite. Also, the many abbreviations (e.g. in chapter 4.1) and awkward formulations in chapter 5 make reading difficult.

**We will revise the mentioned sections to improve the readability, especially with a focus on abbreviations and the fast- and slow-flow concept. The quote "water that is moving slowly through the hydrologic system" is from Kronholm and Capel (2015) and used to explain the concept of slow- and fast-flow.**

The proposed method is built on the IH-UK smoothed minima method following the philosophy of former respective research work performed in Freiburg at the institute of three of the authors under the denomination Wundt/Kille-Demuth method (Demuth, 1989). It is an empirical, statistical approach to only detect and describe the effects of storage in aquifers etc. on streamflow and does not model the hydraulic processes. Baseflow separation methods based on reservoir algorithms are not even mentioned in the present paper though they are the closest to physics and hydraulics.

**A full review of existing baseflow separation methods is beyond the aim. However, we will add a short list of other (incl. reservoir-based) baseflow separation methods and corresponding review papers.**

Line 78 and others: Write Hollick instead of Hollwick.
**Will be corrected.**

Abstract and other places: The authors criticize contemporary "binary" baseflow separation methods "for their arbitrary choice of separation parameters". This is not quite an objective argument. So, like "the DFI is based on characteristic delay curves. . .", other baseflow separation models are calibrated with observed flow recessions and yield good results.
**In order to derive a BFI value many studies use the IH-UK method with the same block size of N=5 or recursive filters with a given parameter a (often a=0.925, cf. Nathan and McMahon, 1990, Eq. 3). We will rephrase, emphasizing that the parameters are most often not adjusted to the hydroclimatology of a specific region.**

The authors probably used data of a number of the same stations as their colleagues in Bern, Switzerland (Meyer et al.2011), who report: "Three different procedures to separate baseflow are applied in 59 catchments in Switzerland. The results show a good coherence of baseflow with well-known storage processes". Why not have a look?
**We were aware of this publication and we like the comparison of different separation methods. Unfortunately, the publication is in German language, but has an English abstract, we will refer to the study in a revised version. Indeed, Meyer et al. highlight considerable differences (Fig. 3) in the derived BFI values depending on the used method (in principle the same outcome as Partington et al., 2012, Fig. 6): All baseflow separation methods are sensitive to the choice of parameter values and therefore hardly comparable unless assessed for the same catchments and record period. For instance, the Demuth method is known to be stricter in separating baseflow and leads consequently to lower BFI values compared to IH-UK or the Wittenberg method (Meyer et al., 2011).**

The authors criticize baseflow separation methods because they "merge different delayed components". Reservoir based separation methods were applied for distinguishing and quantifying different contributions, two examples: Schwarze et al. (1989) created a model of parallel linear reservoirs representing different contributing aquifers or storages. Wittenberg (2003) distinguishes with his nonlinear reservoir method groundwater outflow, groundwater evapotranspiration, abstraction. Is the present method only or particularly suited for regional studies since a linking of flow contributions to catchment characteristics is needed?
**We will rewrite the manuscript to highlight the added value of allowing additional components (in terms of response times). If, as Schwarze et al. (1989) suggested, a modelling approach with two parallel groundwater boxes is more appropriate to simulate flow in catchment, then still the question arise what kind of water will go in which (groundwater) box? A distinct source identification is still not possible even if two instead of one groundwater box is used. We totally agree that differences in the hydrological response of catchments should be reflected in the conceptualization of hydrological models (i.e. variation of model structures for different groundwater flow paths as in Stoelzle et al., 2015). We will reflect on physical/conceptual based modelling approaches (and the mentioned studies above) and the role of aquifer storage in the revised manuscript.**

Line 485: Groundwater recharge does not need saturated soils. Infiltrating water passes the vadose zone via preferential pathways.

**Yes, Reviewer 2 has the same concerns, we will revise the manuscript accordingly.**

Demuth, S. The application of the West German IHP recommendations for the analysis of data from small research basins. Friends in Hydrology, IAHS Publication No. 187, 43-60, 1989

Meyer, R., Schädler, B., Viviroli, D., Weingartner, R. Die Rolle des Basisabflusses bei der Modellierung von Niedrigwasserprozessen in Klimaimpaktstudien. (The role of baseflow in modelling low-flow processes in climate impact studies). HyWa 55,5,244-257, 2011.

Schwarze, R., Grünewald, U., Becker, A., Fröhlich, W. Computer aided analysis of flow recessions and coupled water balance investigations. Friends in Hydrology, IAHS Publication No. 187, 75-83, 1989.

Wittenberg, H. Effects of season and man-made changes on baseflow and flow recession: case studies. HProc 17, 2113-2123, 2003.

**We thank the reviewer for the suggested studies. We will evaluate the value of the suggested references to improve the paper.**

**Additional references:**

**Partington et al. (2012): Evaluation of outputs from automated baseflow separation methods against simulated baseflow from a physically based, surface water-groundwater flow model. Journal of Hydrology, 458, 28–39.**

**Stoelzle et al. (2015): Is there a superior conceptual groundwater model structure for baseflow simulation? Hydrological Processes, 29, 1301-1313.**

---

## Author Comment (AC2) · 9 Aug 2019

**We would like to thank the reviewer for the feedback and the suggestions to improve the manuscript. Here, we respond to each comment (in bold).**

The manuscript presents an interesting idea to distinguish different baseflow components. To my understanding, the main methodological assumptions are correct and could potentially make an interesting contribution for the journal. However, in my opinion, the draft is not well structured and written making it difficult to read, uncertainties regarding the selected dataset and the separation of regimes need to be addressed and the discussion and conclusion should be revised accordingly prior to publication. Below there are some suggestions that might help to improve the manuscript:

**We appreciate the numerous and very helpful comments from Reviewer 2 and will revise the manuscript according to our response below.**

Major comments:
- The overall readability should be improved by favoring short and straight forward formulation: The use of a multitude of abbreviations and the inconsistent usage of wording (e.g. with regard to the term storage) make the paper tough to read: e.g. the introduction needs to be re-written, in my opinion it lacks structure and conciseness; there are many incomprehensive formulations and inconsistencies e.g. the sentence in line 30 to 31 does not make sense to me, are you talking about the magnitude of "sustained streamflow" or more generally of the existence of streamflow? "sustained streamflow and hence freshwater availability" – most freshwater is stored in aquifers; And why does streamflow need to be estimated from BFI? This first general introduction is just very confusing. line 34: What are stored sources? clearly, discharge is coming from "stored sources" whenever it is not raining, the BFI is often interpreted as the contribution from groundwater . . . as you state in line 38. Line 39: you write about water from groundwater, soil and "other delayed source" Which other sources do you mean? Please mention them! There are multiple more examples in the following lines, please try to be more concise in your wording and restructure the introduction!

**Thanks for this comment. We will revise the introduction with explicit focus on consistent use of terms (e.g. storage, sources, delay).**

- The way you report the selection of catchments is critical: you state that human influence on these "headwater" catchments is negligible; (line 254-255): the term headwater catchments is a little misleading for basins of up to 955km2; most of the area in Germany and Switzerland is densely populated, thus human influence might matter: especially when overall magnitudes are small e.g. distinguishing between long delay and baseline delay you will need to make clear that we are not looking at human influence or potential feedbacks from evapotranspiration and vegetation during extended dry periods. The MAG is a regulated basin with huge dams for hydropower and thus highly damped discharge, which makes me a little suspicious if the other selected catchments are suitable for the analysis; please remove MAG and consider double-checking your catchment selection!

**We will remove the term "headwater" and "often negligible" and will describe the catchment and regime characteristics (e.g. also human influences) with more details. We will carefully check all catchments for potential human influences in order to discuss the effect of human influences on the outcome of the analysis.**

- the reasoning for classifying the different regimes, especially when distinguishing between RLWR and RUPR, needs to be further discussed: looking at figures 5 and 6 one could argue that the variation within the groups RLWR and RUPR is larger than the difference between their medians, so from a process point of view (in the end that's what you want to capture) the separation based on mean and max elevation might not be suitable. In Figure 7 you even argue that different elevation classes might be more representative.

**We will investigate the variation within the two rainfall-dominated groups in order to check the reliability of the suggested catchment classification. Our catchment classification has been hypothesis-based, i.e. catchment elevation is a metric to distinguish important drivers of different delayed contributions. We will discuss the value of this approach (see also additional figures below).**

HYBR represents a mixture of snow and rainfall dominated catchments, but obviously as suggest by your results, it is not, can you discuss why?

**We will add more discussion about the specific streamflow response patterns in the HYBR catchments. We will use more information about recession characteristics (as suggested below) to analyze the role of catchment storage in HYBR catchments.**

There might not be an easy solution to these issues, but maybe they can be discussed more detailed. The snow-dominated catchments are significantly smaller than all other catchments (Table 2), please mention that explicitly and update your interpretation accordingly (e.g. line 294 "higher flashiness during summer flows" might be an artefact of catchment size);

**We will highlight that SNOW catchments are in particular smaller than the other study catchments and will revise all statements regarding the flashiness of the catchments.**

maybe you can provide some basic streamflow statistics of the dataset e.g. in Table 1 potentially add magnitude and variation of q5, q50 or recession characteristics with respect to the selected catchment grouping.

**Yes, we will extend Table 1 to present more flow and recession characteristics.**

- also the discussion would benefit from restructuring and improved consistency: e.g. line 409 where can I see "a shift in catchment response" at around 2000m?

**No, we do not see this here, rather with the "2000m" (line 410) we are referring to another study (Pellet and Hauck, 2017). We will make this statement clearer in a revised version. Our reference on Fig. 6 is also wrong in this section (should be Fig. 7). We will rewrite the sentences accordingly.**

line 420: how would you apply the framework worldwide? your case study is on data carrying a strong seasonal signal and elevation gradient. In my opinion the called paradigm shift appears a little too ambitious, as there are (as you also point out in the introduction) several approaches to capture delayed contributions from different storage settings. I don't see how the proposed approach assess (line 439) "different type and number of storages, hence various delayed contributions" While I agree that BFI does not account for single catchment features, also DFI will not identify them specifically (line 445), but you rather get a signal of delayed

outflow from potentially multiple (different) sources. Potentially the climate regime itself might significantly influence Nmax, dry periods in southern Europe or norther latitudes, high-elevation catchments streamflow droughts occur on timescales of < 60 days (up to 4 months). Whereas it might not be relevant for large parts of your study region, it might lead to a biased view on snow-dominated systems and potentially when applying the proposed method elsewhere.

**Thanks for this detailed comment. Review 1 has also raised the question about the transferability of the method to other regions. We have discussed the influence of $N_{max}$ and the number of breakpoints (e.g. line 400, line 396-398, line 475-481). In the revised manuscript we will unite these discussion points and will also highlight that neither BFI nor DFI are able to identify contributing sources in terms of process understanding. DFI instead gives an estimate of the composition of different delayed contributions. Further, the transferability of the method to other regions will be discussed more detailed.**

Also 5.3 starts with a confusing argumentation (lines 483, 484): recharge is crucial everywhere, fair enough, in Alpine catchment seasonal snowpack supplies summer streamflow, however according to table 1 low flow / delayed flow occurs Jan to March, also (line 485) saturated soils are not allowing groundwater recharge. The influence of global warming on melt processes and groundwater recharge is highly depended on the elevation range you are referring to (line 486). To my knowledge it is not yet clear if smaller DB (or smaller groundwater contributions in general) can be directly related to the size of subsurface storages (line 493). There is ongoing discussion if differences in magnitudes are related to variable connectivity of storage and stream, variable precipitation / evapotranspiration in different elevation / exposition or differences in storage recharge. Line 513: If DB is the groundwater contribution, why would less developed soils matter? Again, the ranges you report a quite large, however the SNOW catchments are significantly smaller. The whole argument on storage in SNOW catchments is complicated to follow, you start the argument with Alpine storages are small (but you don't mention who reports that), afterwards you mention numerous studies that report the opposite, to conclude that "high-elevation catchments have larger catchment storage than previously thought".

**Reviewer 1 has comparable concerns about the role of snow, soils and groundwater in alpine catchments. Indeed, the line of argumentation in section 5.3 could be improved. We will discuss potential reasons for large(er) $D_B$-contributions in SNOW catchments and will compare our results with other studies analyzing (dynamic) storage in alpine catchments.**

Some final thoughts on 6: Low streamflow occurrence might be highly variable comparing different years, mainly depending on climate, I'd suggest mentioning that explicitly and re-formulate less definite. Also, the high accordance to elevation gradients might be specific for the Alps, you might not find that in other regions e.g. Scandinavia, southern US;

**We will discuss the role of low flow occurrence during the year, looking at the timing in particular (summer or winter low flow regimes). Indeed, the occurrence is variable comparing different years. The variation is higher for rainfall-dominated than for snowmelt-dominated catchments. We will make clear that the analysis in section 6 is based on a set of generic low flow month for different regime types (i.e. summer low**

**flow, winter low flows) and that in other regions or climates (outside the Alps) other months should be chosen to evaluate the low flow stability.**

Minor comments:

in Figures 1, 3 & 9 the difference between light blue and blue (long vs. baseline) is not visible (in Figure 7 you even replace blue by black, which makes it much more readable, maybe change it also for Figures 1 and 3).
**Will be revised with focus on consistent use of colors.**

the usage of hyphens is quite arbitrary throughout the document, to my knowledge there are clear rules, please check them and change accordingly e.g. line 26 low flow stability index. . . low flow regimes, line 30 groundwater-surface-water-interactions, line 318 5-days… In Figure 1 the dark blue color refers to baseline delay class although it is obviously (the volume) below the baseline, 1b is too small
**Will be revised.**

In line 169: What is the "seasonal low flow period"? How long is it? Where can I see that period of 60 days in the hydrographs of Figure 1a? What exactly is AM, MAM and MQ and how do you calculate them?
**We will explain "low flow period" and the index MAM/MQ with more details. The indices AM, MAM and MQ are explained in line 170-173. However, as details on calculation are missing, we will add more explanation here.**

Line 387: assessed, and may; line 391: sustain low flow for sustained dry periods; line 523: winterly recession;
**Will be changed.**

**Additional material we are considering to include in the revision:**

[Figure]

*Figure 1: (suggestion for additional figure): Correlation strength between the DFI and various flow and catchment characteristics. DFI is calculated for block size N = 1 – 90, the red dot indicates the highest absolute correlation coefficient. With this figure we can argue that a low flow sensitivity measure like MAM/MQ gives better correlation over 60 study catchments than the other 7 variables (e.g. area, slope, elevation etc.). Differences between independent characteristics and flow-derived characteristics should be discussed.*

[Figure]

*Figure 2: A k-means clustering with the 4 relative contributions (short, inter, long, baseline) for all catchments. Hypothesis was that we should find homogenous clusters regarding our classification approach (i.e. homogeneous green, orange, magenta, blue catchment dots in each cluster). Outcomes of cluster analysis should be discussed.*

---

## Author Response (AR1)

**Manuscript id: hess-2019-236**

Dear Editor,

We have revised our manuscript as described in the online responses to reviewer I and II. We specifically addressed the following points to improve the paper focusing on the major concerns of the two reviewers. All changes / revisions can be found in the track change version.

1. We rewrote and improved the **introduction** of the paper, which is now 20% shorter and more concise. Suggested pieces of literature were added, e.g. the Meyer et al. 2011 paper is now integrated into the introduction. The discussion about fast- and slow flow concepts confused the reviewers, we thus concentrate on the main point of the paper: the reasoning why quick- and baseflow separation can be improved with a separation procedure considering multiple delayed contributions. The description of different regime types along the elevation gradient was removed and is now integrated into section 3 "data and regime classification".

2. We now explain more clearly that **the DFI is an extension of the former BFI** procedure and that we see the DFI as a complement to BFI analysis, not as a replacement for BFI.

3. We carried out technical improvements of the paper (e.g. hyphen, abbreviations) to improve the readability of the text. Also, we **revised Fig. 1** with improved coloring and larger subplots (Sect. 2.1).

4. We improved the given **information about the catchments**, explained our indices in more detail, and added metrics about low flow stability and seasonality into Table 1 (Sect. 3). **More catchment, climate and streamflow** characteristics are now shown in Fig. A1 in the appendix. We also revised the suspicious small SNOW catchments (there was an error in Table 1: the mean catchment size of SNOW catchments is 62km$^2$ not 19km$^2$). We also show now that SNOW catchments have a smaller flashiness than HYBR catchments (Fig. A1).

5. We **removed the headwater terminology** completely and state now more clearly that the catchments are not pristine and streamflow might be human-influenced. The potential effects of human influences on the different delayed contributions are now part of the discussion. We also checked that influenced catchments (like the Maggia catchment) are not causing extreme values in the breakpoint estimation or the derived delay classes (Sect. 5.1).

6. Regarding the concerns about our **catchment grouping** based on elevation we performed a **cluster analysis** (new Fig. 9) showing that the grouping by elevation is reasonable as the groups have different patterns of delayed contributions. In addition, we discuss cases where specific catchment or climate characteristics can superimpose the derived patterns based on elevation grouping (Sect. 5.1).

7. Both reviewers had concerns regarding our statements about **groundwater recharge** (Section 5). We rewrote the section and introduced the **concept of dynamic storage** (Staudinger et al., 2017). Dynamic storage can be derived from streamflow analysis. With that, we revised the inconsistent usage of wording (e.g. storage, sustained streamflow, etc.) in the whole paper.

8. Lastly, **we removed Section 6** (Application of DFI: Seasonal sensitivity of low streamflow based on delayed contributions and the corresponding method description in 2.6) as the reviews criticized that the paper was rather long and the application example provided no or only minor additional value on the proposed DFI method. Removing the low flow stability index means also that we have reduced the number of used abbreviations, symbols and indices, which improves the general readability of the paper.

Best regards,

Michael Stölzle

[revised manuscript text omitted]